# BOAD: Discovering Hierarchical Software Engineering Agents via Bandit Optimization

**Iris Xu**[1][*] **Guangtao Zeng**[3]**, Zexue He**[4]**, Charles Jin**[3]**, Aldo Pareja**[3]**, Dan Gutfreund**[2]**,
Chuang Gan**[2,5]**, Zhang-Wei Hong**[2]

## Abstract

Large language models (LLMs) have shown strong reasoning and coding capabilities, yet they struggle to generalize to real-world software engineering (SWE) problems that are long-horizon and out-of-distribution. Existing systems often rely on a single agent to handle the entire workflow—interpreting issues, navigating large codebases, and implementing fixes—within one reasoning chain. Such monolithic designs force the model to retain irrelevant context, leading to spurious correlations and poor generalization. Motivated by how human engineers decompose complex problems, we propose structuring SWE agents as orchestrators coordinating specialized sub-agents for sub-tasks such as localization, editing, and validation. The challenge lies in discovering effective hierarchies automatically: as the number of sub-agents grows, the search space becomes combinatorial, and it is difficult to attribute credit to individual sub-agents within a team. We address these challenges by formulating hierarchy discovery as a multi-armed bandit (MAB) problem, where each arm represents a candidate sub-agent and the reward measures its helpfulness when collaborating with others. This framework, termed Bandit Optimization for Agent Design (BOAD), enables efficient exploration of sub-agent designs under limited evaluation budgets. On SWE-bench-Verified, BOAD outperforms single-agent and manually designed multi-agent systems. On SWE-bench-Live, featuring more recent and out-of-distribution issues, our 36B system ranks second on the leaderboard at the time of evaluation, surpassing larger models such as GPT-4 and Claude. These results demonstrate that automatically discovered hierarchical multi-agent systems significantly improve generalization on challenging long-horizon SWE tasks. Code is available at https://github.com/iamxjy/BOAD-SWE-Agent.

## 1 Introduction

Large language models (LLMs) have achieved remarkable progress in natural language processing [38] and reasoning [21], and are increasingly adopted in solving complex coding problems [58]. Yet solving real-world software engineering (SWE) problems remains challenging [25] for LLMs, particularly for issues that fall outside the training distribution [54]. Despite strong results on SWE-bench-Verified [25], state-of-the-art systems struggle on more recent and out-of-distribution issues in SWE-bench-Live [54].

We hypothesize that one reason current SWE agents generalize poorly is their reliance on a single agent to solve a problem within a single reasoning chain. Solving a software engineering task requires handling multiple sub-tasks—such as locating relevant files, editing code, and running tests—each of which depends only on a subset of information. For example, during code editing, the model only needs to know which line to modify, not how that file was previously located. Retaining such irrelevant context introduces spurious correlations [57; 36], leading models trained on long reasoning chains to overfit to training distributions and degrade on out-of-distribution problems.

Motivated by the need to limit irrelevant context during long reasoning processes, we introduce explicit hierarchical structure to LLM-based agents. This design choice mirrors how human engineers

---

[*]Correspondence: `irisxu@mit.edu`, Massachusetts Institute of Technology[1], MIT-IBM Watson AI Lab[2],
Independent researcher[3], Stanford[4], UMass Amherst[5]

solve complex problems—cognitive science studies show that humans reduce cognitive load by decomposing complex tasks into manageable sub-tasks and integrating their outcomes to form the final solution [32; 37; 33]. Recent work [55] demonstrates that decomposing a complex task into smaller sub-tasks improves LLM performance. In software engineering, such decomposition commonly maps to stages like bug reproduction, fault localization, code modification, and validation [45; 24; 31].

This same idea appears as temporal abstraction in hierarchical reinforcement learning (HRL) [11]: instead of solving problems step by step, the agent delegates control to reusable sub-agents, each defined as a policy for a specific sub-task [40]. An orchestrator coordinates these sub-agents by selecting which one to activate, and each sub-agent runs until it completes its sub-task [40; 19; 10]. Planning at the level of sub-agents shortens the effective decision horizon and abstracts away low-level details, allowing the orchestrator to focus on high-level structure. This abstraction supports generalization because sub-agents capture task-invariant skills—such as repo navigation, information retrieval, or code modification—that can be recomposed across different tasks and contexts, rather than overfitting to task-specific action sequences.

Although hierarchical multi-agent systems for SWE tasks appear promising, prior attempts have achieved only limited success. Existing approaches [8; 34; 14] typically rely on manually constructed systems, where sub-agents are assigned to predefined sub-tasks (e.g., locating faulty files) and coordinated by a human-designed orchestrator. These designs depend heavily on human intuition to decompose tasks and assign agent roles, but such decompositions often fail to align with how LLM-based agents actually reason and behave. Because LLM behavior is difficult to predict, workflows that appear logical to humans can perform poorly in practice (Section 5).

This motivates automating multi-agent design, which removes the need for manual specification and enables the discovery of effective agent designs that may not be obvious to human designers. While prior work has explored optimizing the design of single-agent SWE systems [51], these methods do not address the additional challenges of multi-agent coordination, where agents must interact and share information dynamically. Other approaches optimize fixed, multi-step workflows involving multiple LLM calls [26; 52; 23], but they still rely on predefined pipelines. In contrast, SWE agents operate in open-ended environments where execution order and interactions are not fixed in advance.

Automating multi-agent design introduces two main challenges. First, as the number of sub-agents grows, the design space expands combinatorially, and evaluating each multi-agent design on SWE tasks is far slower than on simpler reasoning benchmarks like multi-hop QA [23]. Second, the evaluation signal is coarse and noisy—success or failure of the multi-agent system reveals little about each sub-agent's contribution. A team of sub-agents may succeed even if one sub-agent consistently makes mistakes and others compensate, making it difficult to assign credit accurately.

To address these challenges, we take a *bottom-up* approach: rather than exhaustively searching over all combinations of sub-agents, we first identify promising sub-agents individually and then compose them into a team. This makes the search space grow linearly instead of exponentially with the number of sub-agents. The key difficulty is credit assignment—a sub-agent's value cannot be measured in isolation since it must collaborate with others to solve an issue (e.g., a localizer alone cannot edit code). Instead of judging success by whether the entire team solves a problem, we estimate each sub-agent's helpfulness, measuring how much it contributes to solving the problem when in combination with other sub-agents. We use LLM-as-a-judge [29] to assess this contribution from the agent's behavior during the problem-solving trajectory, yielding a more informative signal than a binary success or failure outcome.

Once helpfulness is defined, the next question is how to efficiently discover effective sub-agents. Evaluating a sub-agent once is insufficient because its performance depends on its teammates and the task instance, both of which vary stochastically. Yet repeatedly testing all sub-agents would be prohibitively expensive. To balance exploration (trying under-tested sub-agents) and exploitation (focusing on those that have performed well), we formulate multi-agent design as a multi-armed bandit (MAB) problem [17; 1; 20; 50], where each arm corresponds to a sub-agent and the reward is its helpfulness. Well-established exploration algorithms in multi-armed bandits (MAB) [28] provide an effective toolkit for adaptively selecting sub-agents to evaluate and efficiently discovering a set of helpful ones. We refer to this framework as *Bandit Optimization for Agent Design (BOAD)*.

We evaluate our BOAD on SWE-bench-Verified [25], a benchmark for software engineering tasks grounded in real GitHub issues. On SWE-bench-Verified, our method consistently outperforms single-agent systems and manually designed multi-agent systems. On SWE-bench-Live [30], which includes more recently collected issues and presents out-of-distribution challenges, our system ranked second on the leaderboard at the time of evaluation using a 36B model—outperforming larger-scale systems based on Claude and GPT-4. To our best knowledge, our work is the first to show generalization improvements using automatically discovered hierarchical multi-agent systems on challenging long-horizon interactive tasks like SWE-bench.

## 2 RELATED WORKS

Prior works [8; 34; 14] built multi-agent systems for SWE tasks by manually designing the prompts of each sub-agent and the orchestrator that coordinates them. Each sub-agent handled a specific sub-task (e.g., issue localization). While intuitive, these systems required heavy engineering and yielded only marginal gains, leading to limited follow-up work in this direction. Other efforts [30; 53] leveraged diversity across multiple agents or models to improve accuracy through ensembling, where each agent generated a patch and an aggregator selected the final solution. In contrast, we aim to automatically learn a hierarchical multi-agent system—comprising an orchestrator and multiple sub-agents—without manually defining their roles or coordination.

Automating multi-agent design is closely related to prior work on optimizing single-agent systems. Existing approaches have focused on improving an agent's prompt [26; 2; 22] or its scaffold [51], including both tools and prompts, by iteratively refining these components based on task performance. These methods typically optimize a single agent for a given task. In contrast, we study the automated design of multi-agent systems, where different agents are specialized for distinct sub-tasks and must coordinate to solve the overall problem.

Closer to multi-agent systems, several prior works study how to optimize the workflow among agents. In these approaches, a workflow may be represented as a Python function that orchestrates multiple LLM calls Hu et al. [23], or as a graph of agents with distinct roles and prompts [52; 56]. While effective in their respective settings, such workflow optimization methods have seen limited adoption in SWE tasks. One likely reason is that SWE tasks often demand fine-grained, dynamic coordination that evolves as the task unfolds. In contrast to workflows that are largely specified prior to execution, multi-agent systems for SWE typically require agents to adapt and coordinate on the fly in response to intermediate outcomes and newly discovered information.

Prior work has explored automatically generating multi-agent systems [16; 15; 49], but these approaches typically assume a fixed execution scheme, such as a predefined agent ordering [16; 15] or parallel aggregation of agent outputs [49]. In contrast, our work focuses on optimizing how agents coordinate with one another. In addition, existing studies primarily consider simple, inexpensive tasks, whereas we target SWE tasks that involve long-horizon, multi-turn interactions and are costly to evaluate, motivating the need for higher sample efficiency.

## 3 PRELIMINARIES

**Software engineering agents** We study the problem of using LLMs to resolve real-world GitHub issues, where each issue consists of a textual description and a corresponding code repository. Since issues are not self-contained, solving them requires identifying and modifying relevant parts of the codebase. In this work, we focus exclusively on agentic methods [48], where an LLM interacts with a runtime environment through tool use. Such agents can browse files, execute shell commands, run tests, and edit code directly, giving them the flexibility to tackle long-horizon tasks end-to-end.

**Markov Decision Process (MDP)** We model agent–environment interaction as a finite-horizon Markov decision process (MDP) [35], $\mathcal{M} = (\mathcal{S}, \mathcal{A}, r, H)$. At each step $t$, the agent observes a state $s_t \in \mathcal{S}$, consisting of the issue description $x$ and the history of prior tool interactions $h_{t-1} = (a_1, o_1, \ldots, a_{t-1}, o_{t-1})$. The agent samples an action $a_t \in \mathcal{A}$ from its policy $\pi(a_t \mid s_t)$, where $\mathcal{A}$ includes all available tools and commands. Executing $a_t$ yields an observation $o_t \in \mathcal{O}$ (e.g., logs, diffs, or test results), updating the state to $s_{t+1}$. A trajectory $\tau = (s_0, a_0, \ldots, s_T, y)$ terminates at $T \leq H$ when the agent submits a patch $y$ (forced at $T = H$ if none is submitted earlier). Rewards

are sparse:

$$r(s_t, a_t) = 0 \text{ for } t < T, \qquad r(s_T, a_T) = \begin{cases} 1 & \text{if } y \text{ passes all tests,} \\ 0 & \text{otherwise.} \end{cases}$$

The agent's goal is to maximize the expected rewards $J(\pi) = \mathbb{E}_{\tau \sim \pi}[r(s_T, a_T)]$. With sparse rewards and long horizons, discovering successful trajectories is challenging.

**Temporal Abstraction via Semi-MDP (SMDP)**  The Semi-Markov Decision Process (SMDP) framework [39] is widely used to mitigate long-horizon sparse reward problems. Instead of issuing primitive actions $a_t \in \mathcal{A}$, the orchestrator selects a temporally extended action (option) $\omega_t \in \Omega$. Each option corresponds to a sub-agent that executes a sequence of actions $(a_t, \ldots, a_{t+m-1})$ until termination, after which control returns at $s_{t+m}$, where $m$ denotes the duration of a sub-agent. This reduces decision frequency and simplifies planning.

**Multi-Armed Bandit (MAB)**  Multi-armed bandit (MAB) [28] is a special case of an MDP with a single state and no transitions. At each round $t$, the learner selects an arm $a_t \in \mathcal{A}$, receives a stochastic reward $r_t \in [0, 1]$ drawn from an unknown distribution, and seeks to maximize the cumulative reward $\sum_{t=1}^{B} r_t$ over a fixed interaction budget $B$. The MAB framework captures decision-making under uncertainty when only a limited number of trials are available.

## 4 METHOD: BANDIT OPTIMIZATION FOR AGENT DESIGN (BOAD)

Our goal is to automatically discover a set of $K$ sub-agents $\Omega = \{\omega_1, \ldots, \omega_K\}$ together with an orchestrator $\pi$ that maximizes expected task performance (i.e., reward) $r$. A straightforward approach is to directly optimize over both components using evolutionary [13] or random search [12]:

$$\max_{\pi, \Omega} \mathbb{E}_{x \sim \mathcal{D}_{\text{design}}, \tau \sim \pi} [r(s_T, a_T)], \tag{1}$$

where $\mathcal{D}_{\text{design}}$ denotes a *design set* of representative problem instances. However, evaluating a candidate pair $(\pi, \Omega)$ requires executing full interaction trajectories $\tau$ and repeatedly querying the reward function $r$. In software engineering (SWE) tasks, such evaluations are long-horizon, multi-turn, and computationally expensive, making naive joint search over $(\pi, \Omega)$ impractical.

**Agent design as a multi-armed bandit.** To address this challenge, we reformulate agent discovery as a multi-armed bandit (MAB) problem [28]. In this formulation, each arm corresponds to a candidate sub-agent $\omega_i$, and at each round $t$, a subset of $K$ sub-agents is selected and evaluated under an orchestrator. Casting agent design as a bandit problem allows us to leverage principled exploration–exploitation strategies [27; 9]: these algorithms can allocate more evaluations to promising sub-agents while still exploring alternatives. This substantially reduces wasted computation on poor designs and makes automatic discovery of multi-agent systems feasible under the high evaluation costs characteristic of SWE tasks. We present the formal bandit formulation in Section 4.1. Towards this end, we must solve the following problems:

1. The space of possible orchestrators and sub-agent sets is vast and initially unknown, making it infeasible to pre-enumerate arms, as required by standard bandit algorithms.

2. Even if we evaluate a sub-agent along with an orchestrator and the other sub-agents, credit assignment is ambiguous: some sub-agents may succeed only by "free-riding" on others, so the observed reward does not necessarily reflect their individual contribution.

Next, we illustrate how we tackle these challenges in the following sections.

### 4.1 AGENT DESIGN AS A MULTI-ARMED BANDIT PROBLEM

The space of orchestrator–subagent pairs $(\pi, \Omega)$ is vast and infeasible to enumerate. To make the search tractable, we maintain an archive $\Gamma$ of candidate sub-agents. Instead of treating each subset of sub-agents $\Omega$ as an arm, we treat each sub-agent $\omega \in \Gamma$ as an arm, enabling information sharing across different sub-agents (will be explained below). At each round $t$, the algorithm selects a subset $\Omega_t \subseteq \Gamma$ by choosing $K$ arms, instantiates an orchestrator $\pi_t$ for $\Omega_t$, evaluates $(\pi_t, \Omega_t)$ on example

problems from a *design* set, and propagates feedback to all participating sub-agents. Feedback is calculated for each sub-agent independently to tackle the credit assignment problem (details in 4.2). Because sub-agents appear in multiple subsets, each evaluation, even if unsuccessful, provides signal for multiple subsets at the same time. This formulation supports efficient credit assignment, reduces redundant exploration, and progressively refines estimates $u_\omega$ for each $\omega \in \Gamma$. Algorithm 1 summarizes the procedure, with further details provided below.

**Bootstrapping a sub-agent archive**  We begin with an initial archive $\Gamma_0$ of candidate sub-agents. This archive is generated by prompting an LLM with the template in Appendix A.1.2. However, simply generating sub-agents is insufficient because the orchestrator may not know how to invoke them. We present sub-agents as tools to the orchestrator and adopt the standard tool-calling protocol from SWE-agent [48]. To call a sub-agent, the orchestrator must parse its docstring to understand the functionality and supply the required inputs. For example, an issue-localizer sub-agent requires the issue summary as input; without it, the sub-agent cannot operate. To ensure this, we introduce a warm-up stage that rewrites each generated sub-agent's docstring into a precise specification of its inputs and outputs, enabling the orchestrator to integrate it correctly. Details are provided in Appendix A.1.1.

**Sub-agent evaluation**  At each round $t$, we select a set of $K$ sub-agents $\Omega_t = \{\omega_1, \ldots, \omega_K\} \subseteq \Gamma_{t-1}$. Given this set, an orchestrator $\pi_t$ is instantiated by prompting an LLM (see Appendix A.1.3), and the system $(\pi_t, \Omega_t)$ is evaluated on a subset of example problems from a design set. This evaluation yields a performance score $u_\omega \in [0, 1]$ for each sub-agent $\omega \in \Omega_t$. A straightforward choice for $u_\omega$ is the success rate of the system on the design set, but as discussed in Section 4.2, this metric is suboptimal and we propose a more effective alternative.

**Balancing exploration and exploitation on sub-agent selection**  After bootstrapping the archive, the next challenge is deciding which sub-agents to evaluate in each round. To balance exploration with exploitation, we adopt the Upper Confidence Bound (UCB) [28] strategy. For each sub-agent $\omega \in \Gamma_{t-1}$, we track its empirical mean $\hat{\mu}_\omega(t)$ of the performance score of $u_\omega$ and selection count $n_\omega(t)$ up to round $t$. The UCB score of a sub-agent $\omega$ at round $t$ is defined as

$$\text{UCB}_\omega(t) = \hat{\mu}_\omega(t) + \sqrt{\frac{2 \ln t}{n_\omega(t)}}.$$

The first term favors sub-agents with high observed performance, while the second term gives an optimism bonus to under-sampled sub-agents (i.e., exploration). At each round $t$, we select the top-$K$ sub-agents based on their UCB scores, ensuring that evaluations increasingly focus on strong candidates while still allocating time to uncertain ones.

**Expanding the archive**  A fixed archive risks stagnation: once UCB identifies a few strong sub-agents, it will repeatedly exploit them, leaving little opportunity to discover new behaviors. To address this, we expand the archive dynamically using a Chinese Restaurant Process (CRP) [3; 44]. At each round $t$, we prompt the LLM to generate a new sub-agent distinct from those in the current archive $\Gamma_{t-1}$ (see Appendix A.1.2). The probability of introducing a new sub-agent is

$$\Pr(\text{new at } t) = \frac{\theta}{\theta + |\Gamma_{t-1}|},$$

where $\theta > 0$ is a concentration parameter. This mechanism ensures diversity: when the archive is small, new sub-agents are frequently added; as the archive grows, the probability decreases, shifting the emphasis toward reuse of existing ones. Over time, the expected number of distinct sub-agents after $T$ rounds grows as $O(\theta \log T)$, providing unbounded but controlled expansion. We also run the warmup stage (Appendix A.1.1) to ensure the sub-agent is usable by the orchestrator.

## 4.2 HINDSIGHT CREDIT ASSIGNMENT

A central challenge in our framework is defining the performance score $u_\omega$ of individual sub-agents $\omega$. A simple approach is to set the score of a sub-agent to the success rate of all trajectories that

---

**Algorithm 1** Bandit Optimization for Agent Design (BOAD)

---

**Require:** budget $B$, number of sub-agents to select $K$, concentration $\theta$
 1: Initialize archive $\Gamma_0 \leftarrow$ BOOTSTRAP.
 2: **for** $t = 1, 2, \ldots, B$ **do**
 3:      With probability $\frac{\theta}{\theta+|\Gamma_{t-1}|}$, create a new sub-agent $\omega_{\text{new}}$ and set $\Gamma_t \leftarrow \Gamma_{t-1} \cup \{\omega_{\text{new}}\}$;
      otherwise set $\Gamma_t \leftarrow \Gamma_{t-1}$.
 4:      **for** each $\omega \in \Gamma_t$ **do**
 5:          **if** $n_\omega(t-1) = 0$ **then**
 6:              $\text{UCB}_\omega(t) \leftarrow +\infty$                              ▷ force initial exploration
 7:          **else**
 8:              $\text{UCB}_\omega(t) \leftarrow \hat{\mu}_\omega(t-1) + \sqrt{\frac{2\ln t}{n_\omega(t-1)}}$
 9:          **end if**
10:      **end for**
11:      Select top-$K$ sub-agents based on UCB scores as a set of sub-agents $\Omega_t$.
12:      Instantiate orchestrator $\pi_t$ conditioned on $\Omega_t$.
13:      Evaluate $(\pi_t, \Omega_t)$ on a subset of training problems; observe performance score $u_\omega \in [0, 1]$
      (Sec. 4.2).
14:      Update $\hat{\mu}_\omega(t)$ and $n_\omega(t)$ for each $\omega \in \Omega_t$.
15: **end for**

---

include it:

$$u_\omega = \frac{1}{|\mathcal{T}_\omega^t|} \sum_{\tau \in \mathcal{T}_\omega^t} \mathbb{1}\{\tau \text{ is successful}\},$$

where $\mathcal{T}_\omega^t$ is the set of trajectories at round $t$ in which $\omega$ is used by the orchestrator $\pi$. However, this suffers from a "free-rider" problem: a sub-agent may appear effective simply because it often co-occurs with strong sub-agents, even if it contributes little itself.

To overcome this, we adopt a hindsight-based credit assignment strategy. The idea is to reward a sub-agent whenever its actions help the orchestrator make progress toward solving the problem, even if the orchestrator ultimately fails. Thus, sub-agents that provide useful intermediate steps are credited, while those that do not are penalized, regardless of the outcomes. Concretely, let $\tau = (a_1, o_1, \ldots, a_T, o_T)$ denote the trajectory of actions and observations produced during problem solving. For each sub-agent $\omega$ that appears in $\tau$, we query an LLM judge (Appendix A.1.4) with the trajectory and obtain a binary label $\ell_\omega(\tau) \in \{0, 1\}$, where $\ell_\omega(\tau) = 1$ indicates that the LLM judge deems $\omega$'s contribution in the trajectory as helpful. The performance score of sub-agent $\omega$ is then defined as the empirical average over all evaluated trajectories:

$$u_\omega = \frac{1}{|\mathcal{T}_\omega^t|} \sum_{\tau \in \mathcal{T}_\omega^t} \ell_\omega(\tau).$$

This hindsight-based score $u_\omega \in [0, 1]$ provides a more reliable estimate of the utility of $\omega$ than success rates. By directly linking credit to judged contributions, it avoids free-riding effects.

## 5 EXPERIMENTS

Our experiments address the central question: *Can properly designed hierarchical multi-agent systems improve the generalization performance of SWE agents?* We further analyze how the systems discovered by our algorithm differ from human-designed ones and examine the contribution of each design choice to the overall performance gains.

### 5.1 SETUP

**Task format and datasets.** We evaluate on the SWE-BENCH benchmarks: SWE-BENCH VERIFIED (500 instances) [25] and SWE-BENCH LIVE (300 instances) [54]. VERIFIED is a curated, frozen set of real GitHub issues, while LIVE continuously adds newly collected, human-verified issues from active repositories, making it more resistant to data contamination [46] and better suited for

testing generalization to out-of-distribution problems. Each instance includes a GitHub issue, a repository-specific container image, and an executable test harness. The agent must interact with the repository (files and, when available, history) and produce a patch that resolves the issue by passing *all* tests (`pass-to-pass` and `fail-to-pass`).

To avoid overfitting and limit design-time compute, we construct a small yet diverse *design set* by sampling one random issue per repository (12 total) from VERIFIED, which we find to be sufficient based on an ablation over design-set size (A.3.1). The design set is disjoint from all issues in LIVE. All results in Tables 2 and 3 are reported on VERIFIED and LIVE (lite) splits. We also report the result of BOAD on VERIFIED (HELD OUT) that exclude the 12 issues used in the design set.

**Optimization Details** During BOAD optimization, we run $B = 20$ rounds of the bandit loop (Algorithm 1) (testing on up to 100 rounds shows that sub-agents created after around 20 rounds are generally worse, and 20 iterations is enough to converge to a state where helpful rate and UCB rankings align). Each time a new sub-agent is created, the sub-agent first goes through a warm-up process, which uses randomly sampled instances from the design set to iteratively refine the documentation/instance prompt of the sub-agent, ensuring that the sub-agent is usable. We use $W = 4$ rounds for the warm-up process. Each round samples $K = 3$ sub-agents and evaluates them on the design set. The optimization took around 12 hours on a machine with 56 CPU cores and 440GB RAM. For running SWE-agent (orchestrator and sub-agent LLM inference), we deploy Seed-OSS-36B-Instruct on one H100 GPU node. Note that by the tenth iteration, the top-2 most helpful subagents are the same as the ones converged on later. Running $B = 10$ iterations took less than 7 hours.

**Implementation.** All experiments use the SWE-AGENT scaffold with a set of default tools from SWE-agent [48]: `edit_anthropic` (file viewing/editing), `bash` (restricted shell commands), and `submit`. The orchestrator calls sub-agents through the same XML-based tool-calling API, passing information via a *context* parameter; sub-agents return outputs through this channel without access to the orchestrator's history. We use Claude-4 to generate candidate designs (Section 4.1) with temperature 0.0 and evaluate sub-agent helpfulness (Section 4.2). For execution, both orchestrator and sub-agents use Seed-OSS-36B-Instruct with temperature 0.0, unless specified, a strong instruction-following model that is not heavily tuned on SWE tasks. This choice ensures improvements reflect the benefit of orchestration rather than fine-tuning on SWE-task specific data. Each sub-agent is equipped with prompts discovered by BOAD, defined with docstrings and argument specs, and given access to the same default tools as the orchestrator.

Finally, for evaluation, we use the two subagents with the highest helpfulness score (the hindsight-based score) found during the subagent discovery (Appendix A.2.1). We also ablate by varying k in the top-k selection and by using success rate instead of helpfulness as the ranking metric (Section 5.3).

## 5.2 MAIN RESULTS

**Success Rate** Table 1 shows that with Seed-OSS-36B-Instruct, BOAD resolves **20.0%** of issues on LIVE, ranking second on the leaderboard at the time of evaluation and outperforming larger models in popular scaffolds (e.g., GPT-4o, DeepSeek-V3, GLM-4.5-Air, Claude 3.7 Sonnet). This is a **63%** improvement over the same model with default SWE-agent tools. On VERIFIED, BOAD achieves **53.12%**, surpassing many larger models (e.g., GPT-4o, Claude 3.5 Sonnet OpenHands, DeepSeek-R1, DeepSeek-V3) and setting a new state of the art among smaller models (e.g., Qwen3-Coder-30B-A3B-Instruct, Devstral-Small-2505), with a **13.4%** gain over the default SWE-agent. Interestingly, adding manually designed sub-agents (Appendix A.2.4) from prior work [8; 34; 14] lowers performance, indicating that human-crafted roles can be misaligned with LLM behavior. Overall, these results demonstrate that BOAD automatically discovers orchestrator–sub-agent structures that not only boost in-distribution performance but also generalize more effectively to out-of-distribution tasks.

**Comparison with Evolutionary Baseline** We also implement an evolutionary search for a multi-agent system adapted from Automated Design of Agentic Systems [23], as a baseline to compare BOAD against. Implementation details for the evolutionary search are provided in Appendix A.4.1. When using the same number of evaluation instances (i.e., number of SWE-Bench patches generated), we find that the sub-agents generated by the evolutionary search achieve worse performance (17.0%

vs 20.0% on SWE-Bench-Live) than BOAD. Additionally, for the same number of iterations, the cost of Claude API calls is more than double that of BOAD due to the need of generating many sub-agents at each iteration, whereas BOAD reuses subagents across iterations.

Table 1: Success rate on SWE-BENCH VERIFIED and SWE-BENCH LIVE.

| Scale | Model | Scaffold | Verified Resolved (%) | Live Resolved (%) |
|-------|-------|----------|----------------------|-------------------|
| Large | GPT-4o [48] | SWE-agent | 23.0 | 10.0 |
| | GPT-4o [47] | Agentless | 38.8 | 11.7 |
| | Claude 3.5 Sonnet [5] | Agentless | 50.8 | – |
| | Claude 3.5 Sonnet [5] | OpenHands | 53.0 | – |
| | Claude 3.7 Sonnet [4] | SWE-agent | 62.4 | 13.7[1] |
| | Claude 4.0 Sonnet [6] | SWE-agent | 66.8 | – |
| | Claude 4.0 Sonnet [6] | OpenHands | 70.4 | – |
| | DeepSeek-R1 [21] | Agentless | 49.2 | – |
| | DeepSeek-V3 [18] | Agentless | 42.0 | 13.3 |
| | GLM-4.5-Air [41] | OpenHands | 57.6 | – |
| | GLM-4.5-Air [41] | SWE-Agent | – | 17.7 |
| | Qwen3-Coder 480B/A35B Instruct [43] | OpenHands | 69.6 | 24.7 |
| Small | Qwen3-Coder-30B-A3B-Instruct [43] | SWE-agent | – | 17.0 |
| | Qwen3-Coder-30B-A3B-Instruct [43] | OpenHands | 51.6 | – |
| | Devstral-Small-2505 [7] | OpenHands | 46.8 | – |
| | Seed-OSS-36B-Instruct [42] | SWE-agent (baseline) | 49.8 | 12.3 |
| | Seed-OSS-36B-Instruct [42] | SWE-agent + Manual Sub-agent | 47.4 | 14.0 |
| | Seed-OSS-36B-Instruct [42] | SWE-agent + Evolutionary Search | 46.0 | 17.0 |
| | Seed-OSS-36B-Instruct [42] | SWE-agent + **BOAD** | **53.2**[2] | **20.0** |

**Token Analysis**   In addition to success rate, we analyze the test-time token usage of the hierarchical multi-agent system discovered by BOAD in comparison to the default single-agent system. Hierarchical multi-agent systems introduce communication overhead, since agents must exchange information, but they can also reduce context length: sub-agents focus on specialized sub-tasks while the orchestrator handles high-level coordination without low-level details. Table 2 compares token usage between SWE-agent and BOAD. Total tokens refer to the average sum of input and output tokens per issue, while max input tokens capture the average maximum input length per instance. Surprisingly, the total token count is comparable—and even lower on SWE-bench-live—than in the original SWE-agent. Moreover, BOAD consistently reduces input tokens, confirming that task decomposition shortens context length.

Table 2: **Token usage.** BOAD lowers input token counts, thus shortening the model's input context length.

| Metric | Setting | Verified | Live |
|--------|---------|----------|------|
| Total tokens (M) | SWE-agent | 0.92 | 1.49 |
| | SWE-agent + BOAD | 0.93 (+0.7%) | 1.13 (-23.8%) |
| Max input tokens | SWE-agent | 34.6k | 49.0k |
| | SWE-agent+BOAD | 30.5k (-11.6%) | 36.7k (-25.0%) |

## 5.3   ABLATION STUDIES AND ANALYSIS

**Does prompt optimization explain the gains?** One possible explanation for BOAD 's performance improvement is that it simply arises from better prompt optimization of SWE-agent. To test this, we introduce a baseline that optimizes the SWE-agent prompt without adding sub-agents (w/o Sub-agent). We run 10 iterations: in each, a new SWE-agent prompt is generated by prompting Claude-4 with the template shown in A.1.5, evaluated on 12 issues (the same setting as BOAD). The first iteration is initialized without history, and from the second onward, prompt generation is conditioned on the top five prompts from previous rounds, ranked by performance. Results in Table 3 show that prompt optimization alone does not reach the performance of BOAD, indicating that the gains are not solely due to prompt tuning but from the discovery of effective sub-agents and orchestration.

---

[1]The SWE-bench-live leaderboard score was 17.7, based on an earlier issue set from April 2025.

[2]53.1 on the SWE-bench-verified set excluding the 12 issues used in the design set.

Table 3: **Ablation studies and analysis.** Each row corresponds to one research question. Results are reported on SWE-bench Live using Seed-OSS-36B-Instruct unless otherwise specified.

| Research Question | Configuration | SWE-Bench Live (%) |
|---|---|---|
| Does prompt optimization explain the gains? | w/o Sub-agent | 16.3 |
| | w Sub-agent | **20.0** |
| Do more sub-agents improve performance? | Top-5 sub-agents | 13.7 |
| | Top-4 sub-agents | 16.7 |
| | Top-3 sub-agents | 16.3 |
| | Top-2 sub-agents | **20.0** |
| | Top-1 sub-agent | 16.3 |
| Do we need to customize the orchestrator? | w/o customization | 16.7 |
| | w customization | **20.0** |
| Is expanding the sub-agent archive needed? | w/o expansion | 17.0 |
| | w expansion | **20.0** |
| Is hindsight credit assignment necessary? | Top-3 subagents (success rate) | 11.3 |
| | Top-3 subagents (helpfulness) | **16.3** |
| | Top-2 subagents (success rate) | 15.3 |
| | Top-2 subagents (helpfulness) | **20.0** |
| Are discovered sub-agents transferable to other models? | Claude 3.7 Sonnet | 13.7 |
| | + Top-2 sub-agents (helpfulness) | **16.3** |

**Do more sub-agents improve performance?** One might expect performance to improve as more sub-agents are added, since each can specialize. To test this, we vary the number of top-$K$ sub-agents from 1 to 5 based on the helpfulness score (Section 4.2) and evaluate on LIVE. Surprisingly, Table 3 shows that performance peaks with exactly two sub-agents, achieving 60/300 (20.0%). A single sub-agent (49/300) fails to leverage specialization, while larger teams of three (49/300), four (50/300), or five (41/300) reduce performance due to communication and coordination overhead. These results suggest that small, focused teams strike the best balance, outperforming both minimal and overly large teams of sub-agents.

**Do we need to customize the orchestrator?** We next ask whether gains come solely from sub-agent discovery or if the orchestrator must also adapt to its team. We compare two prompting strategies: (i) a generic prompt encouraging sub-agent calls (Appendix A.1.1), and (ii) a customized prompt generated by Claude-4 that explicitly references the top two sub-agents (selected by helpfulness scores) and outlines a plan for using them. Both settings use the same sub-agent set, but only the customized prompt allows the orchestrator to reason about and plan calls to specific sub-agents. Results in Table 3 show the customized orchestrator (w customization) achieves 60/300 (20.0%), versus 50/300 (16.7%) (w/o customization) for the generic one. This indicates that while sub-agents provide new capabilities, the orchestrator must also be specialized to effectively coordinate them.

**Is expanding the sub-agent archive needed?** As discussed in Section 4.1, the initial archive may be limited, and adding new sub-agents during the design process could be necessary to discover stronger ones. To test this, we compare orchestrator performance using (i) sub-agents from the initial archive (w/o expansion) and (ii) sub-agents selected at the end of the design process (w expansion). Both settings use two sub-agents, consistent with our best configuration in Section 5.2, and the orchestrator is generated as described in Section 4.1. Results in Table 3 show that final sub-agents outperform those from the initial archive, highlighting the importance of expanding the archive over time.

**Is hindsight credit assignment necessary?** To address free-riding issues in sub-agent selection (Section 4.1), we use a helpfulness score to measure each sub-agent's contribution. To test its importance, we compare orchestrator performance when sub-agents are selected by (i) individual success rate versus (ii) helpfulness score. As shown in Table 3, helpfulness-based selection consistently outperforms success-rate selection, indicating that hindsight credit assignment (Section 4.2) is essential for identifying useful sub-agents.

**Are discovered sub-agents transferable to other models?** Since BOAD optimizes sub-agents for a specific model, we ask whether the best sub-agents differ across models and whether effective

sub-agents can transfer. To test this, we apply the sub-agents from Section 5.2 to SWE-agent+Claude-3.7-Sonnet. As shown in Table 3, the discovered sub-agents do transfer to some extent, though the gains are smaller than those achieved with Seed-OSS-36B-Instruct, the model used for sub-agent optimization.

### 5.4 QUALITATIVE ANALYSIS OF SINGLE- VS MULTI-AGENT OUTCOMES

We manually inspected trajectories in which the single- and multi-agent systems produced different outcomes. Three recurring patterns emerged:

1. **Over-editing (multi-agent advantage).** Single agents frequently produced extremely long patches, including attempts to create new tests and edits outside the scope of the bug. Such patches inflate apply time and increase the chance of failing `pass-to-pass`, even if the agent is able to address the primary fault. In contrast, the multi-agent system tended to emit short, localized patches, highlighting the advantage of separating phases like localization and editing/testing.

2. **Multi-site fixes and coverage (multi-agent advantage).** When the fix required edits at multiple call sites or modules, the single agent often either over-edited unrelated regions or missed one or more necessary locations. The hierarchical system mitigated both omission and extraneous edits by delegating to a sub-agent for localization, which did a thorough analysis of the repository before making any targeted modifications.

3. **Error propagation from unvalidated sub-agent outputs (multi-agent failure mode).** In a minority of cases, the multi-agent system failed while the single agent succeeded. Inspecting these outputs, we found that erroneous sub-agent outputs (e.g., incomplete span identification, misinterpretation of the issue) were accepted as ground truth by the orchestrator, leading subsequent steps astray. Because there is no intermediate validation or self-checking, the orchestrator has limited ability to recover from such upstream errors.

These observations align with our hypothesis: hierarchical delegation constrains edit scope and improves coverage of multi-site fixes, but introduces a new dependency on the *quality of sub-agent handoffs*. Incorporating lightweight verification (e.g., span cross-checks, invariant tests, or dual-read localization) is a promising mitigation for the third failure mode.

## 6 DISCUSSION & CONCLUSION

We present BOAD, a framework that formulates hierarchical multi-agent design as a sequential, online decision making problem to automatically discover multi-agent systems for long-horizon software engineering tasks. Our experiments show that automatically discovered sub-agents, when combined with a customized orchestrator, outperform single-agent and manually designed multi-agent systems on both SWE-BENCH VERIFIED and SWE-BENCH LIVE.

**Limitations and Future Work:** We find that discovered sub-agents transfer across models only partially and failure cases highlight error propagation when orchestrators unconditionally accept sub-agent outputs. Future work should explore evolution on large models, adaptive team sizing, verification, as well as extending the framework to domains beyond software engineering.

### ACKNOWLEDGMENT

We thank members of the Improbable AI Lab at MIT and MIT-IBM Watson AI Lab for helpful discussions and feedback.

### AUTHOR CONTRIBUTION

- **Iris Xu**: Led the project, implemented and designed the core algorithm, conducted all the experiments, and wrote the experiment section.
- **Guangtao Zeng**: Implemented the baselines, explored the preliminary design of the ideas of this work, and wrote the related work section.

- **Zexue He**: Explored the preliminary design of the ideas of this work and helped revise the manuscript.
- **Charles Jin**: Assisted and supported the implementation of our method.
- **Aldo Pareja**: Assisted and supported writing.
- **Dan Gutfreund**: Coordinated the compute and infrastructure.
- **Chuang Gan**: Coordinated the compute and infrastructure.
- **Zhang-Wei Hong**: Led the writing and ideation of the project, coordinated the team and resources, and explored the preliminary design of the ideas of this work.

## ETHICS STATEMENT

Coding agents hold strong promise for automating code generation and bug fixing, but they also carry risks of unintended or harmful outputs. For instance, an LLM-based agent may produce commands that could compromise a system (e.g., downloading unauthorized packages or deleting user files with `rm -rf`). To mitigate these risks in our study, all experiments were conducted within Docker containers, providing isolated and sandboxed environments that prevent harmful commands from impacting real user devices and substantially reducing the potential for actual harm. Our implementation also builds upon the SWE-Agent framework, which has been previously published and reviewed under established ethical standards. We carefully follow its curated protocols and licensing requirements.

Nevertheless, as with any AI-based coding agent framework, there remains the risk of deliberate misuse, for example, a malicious user prompting the system to generate harmful or hacking code. While our contribution is centered on advancing the technical design of hierarchical coding agents, we emphasize that real-world deployment should be coupled with responsible auditing and oversight, so that potential misuse and unintended consequences can be effectively mitigated.

## REPRODUCIBILITY STATEMENT

For all open-source LLMs (e.g., Seed-OSS-36B-Instruct, Qwen3-Coder-30B-A3B-Instruct), we rely on their official releases. Commercial LLMs and LLM-based tools are accessed through their official APIs and reference implementations. We provide detailed implementation notes of BOAD, including the prompt design for meta-agents, sub-agents, and LLM judges, in Section 5.1. To further support reproducibility and foster future research, we will also release all of our code, used data, and prompts.

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

# A EXPERIMENT DETAILS

## A.1 PROMPT TEMPLATES

### A.1.1 PROMPT TEMPLATE FOR REFINING SUB-AGENT DURING WARMUP STAGE

---

**Prompt template for refining subagent during warmup stage**

```
You are improving a subagent's prompts/config for a software engineering (SWE) automation system, based
↪  on recent run trajectories. The subagent is used by an AI main agent to address code issues.

CONTEXT
- You will receive trajectory summaries below, starting with the main agent's trajectory, followed by
↪  any subagent trajectories in call order.
- Each summary shows what the agent did, what was observed, and how far it progressed.

GOAL
Analyze the subagent's performance and suggest improvements to make it:
1. More discoverable by the main agent (when appropriate)
2. More reliable in its behavior
3. More useful in its output

ANALYSIS FRAMEWORK
Consider these questions:
- Did the main agent discover and use the subagent when it should have?
- Did the subagent behave as expected and return useful information?
- Were there missed opportunities or inefficient behaviors?

IMPROVEMENT TYPES
Focus on one or more of these areas:
1. **docstring**: Make the subagent easier for the main agent to discover and choose appropriately.
↪  CRITICAL: Make sure to include "[subagent]" at the beginning of the docstring.
2. **context_description**: Improve the description of the 'context' argument (the only argument) to be
↪  clearer and more helpful
3. **instance_template**: Better incorporation of context, clearer framing for each problem instance

Note that the docstring and context_description are visible only to the main agent, while the
↪  instance_template is visible only to the subagent.
Thus, if the subagent was not called, you should not edit instance_template. Similarly, if the only
↪  issue is the subagent's trajectory, not how it was called, do not edit docstring or
↪  context_description.

PRINCIPLES
- Make surgical, targeted improvements rather than broad rewrites
- Preserve existing style and capabilities. Only edit components that need improvement.
- Focus on clarity, discoverability, and reliability
- Ensure generality. Avoid repo- or issue-specific assumptions.
- CRITICAL: DO NOT WRITE ANYTHING SPECIFIC TO THE PARTICULAR CODEBASE, PROJECT, OR DOMAIN.

OUTPUT FORMAT
First, explain your reasoning about what issues you noticed with the provided trajectory and what
↪  improvements you're making. Then, output the YAML in a code block.
**IMPORTANT**: Only suggest edits when you identify a clear, specific problem. If the entire subagent
↪  is working well, use an empty updates dictionary.

Sample outputs:
**When improvements are needed:**
Explain your reasoning here.
```yaml
updates:
  docstring: "<improved docstring if needed>"
  context_description: "<improved context argument description if needed>"
  instance_template: "<improved instance template if needed>"
```

RULES
- Only include keys that you intend to change.
- Start with `updates:` as the top-level key. If there are no updates to make, the value for `updates`
↪  should be an empty dictionary.
- You may update any combination of the three fields (docstring, context_description,
↪  instance_template), but only include a field if it needs improvement.
- No explanations or extra content in the YAML
- Keep each field concise but complete

HEURISTICS
- **Discovery issues**: Strengthen docstring with clear use cases and when to invoke
- **Insufficient context passed to subagent**: Improve context_description with clearer argument
↪  explanation
- **Subagent behavior and output (Incorrect subagent trajectory or return information)**: Improve
↪  instance_template with better instructions and output specifications

{{TRAJECTORIES}}
```

---

### A.1.2 META AGENT PROMPTS

---

**Prompt for generating a new subagent configuration**

```
You are an expert at designing custom tools for SWE-agent, an autonomous agent that can resolve code
↪  issues in large repositories.

YOUR TASK
Invent a subagent tool for SWE-agent.
- The subagent should enable the main agent to better perform its task of automatically resolving code
↪  issues in large static repositories.
- Design for broad applicability across the full workflow. Create broad subagents that are can solve an
↪  entire step of the pipeline, such as:
  - code localization
  - reproducing issues and running scripts/tests
  - code editing/patching
  - code testing
- The subagents created should ONLY be focused on correctness of the final patch (e.g. style,
↪  complexity of code does not matter)
- PRIORITIZE TOKEN EFFICIENCY: Design concise, focused subagents that use minimal tokens. Avoid verbose
↪  explanations or redundant information that wastes tokens.
- If you see a subagent that looks good but has bad token efficieny, you may generate a similar
↪  subagent with the same function but better implementation.
- The subagent takes a SINGLE argument that is a string, called "context".
- BE NOVEL! Think carefully about how to help the main agent perform one of its subtasks.
  - Example subagents include localize, patch_editor, or code_tester.
- Do not create a subagent that overlaps with previous subagents (other than the token efficiency
↪  situation).
- In your reasoning, you must explicitly list which steps the subagent supports (examples: explore,
↪  read/search, edit, run, validate) and the expected outputs for each supported step.
  - CRITICAL: Output exactly ONE YAML document with the tool under a single key, which is the name of
  ↪  the tool.
  - The name of the tool should be simple and descriptive.
  - The docstring for each tool should be comprehensive and describe what the output contains, as well
  ↪  as the state of the repository after the subagent is finished (if files will be edited or not,
  ↪  etc.).

The structure must be exactly as follows:
Add reasoning here about WHY you design this subgaent...
```yaml
tool_name:
  signature: "tool_name <context>"
  docstring: docstring: "comprehensive description of this subagent, the output, and the state of
  ↪  repository on completion. Starts with '[subagent]:"
  arguments:
    - name: context
      type: string
      description: "detailed description of what the context parameter should contain"
      required: true
  subagent: true
```

Sample output:
It may be useful to have a patch editor subagent. This would go well with previous subagents and help
↪  the main agent more efficiently patch the issue.
```yaml
patch_editor:
  signature: "patch_editor <context>"
  docstring: "[subagent] Fixes a specific part of code that has errors. Outputs the changes made with
  ↪  reasoning. After calling, the correct changes are already implemented in the repository."
  arguments:
    - name: context
      type: string
      description: "A string containing the specific file path to make edits in, the lines where edits
      ↪  need to be made, a comprehensive description of the issue with the code (do not assume the
      ↪  subagent has any information about the repository or problem statement), and what to edit."
      required: true
  subagent: true
```
{{PREVIOUS_ITERATION_FEEBACK}}
```

---

**Prompt for generating subagent templates**

```
You are an expert at creating SWE-agent subagent configuration files for automating code-patching tasks
↪  in large GitHub repositories.
Given a description of the subagent, you need to generate the system_template and instance_template
↪  parts that will be used in the subagent configuration.

IMPORTANT FORMATTING RULES:
- First output your reasoning that details your thinking process for creating the templates. Then,
↪  output a yaml block with both templates.
- Use MINIMAL spacing - avoid excessive blank lines
- Use only SINGLE blank lines between sections (never double or triple spacing)
- Keep templates compact and readable without unnecessary whitespace
- CRITICAL: Use YAML literal block syntax with | and |- (see example below)
- Do NOT use quoted strings - use literal blocks to avoid quotes in output
```

```
- Replace ONLY text in [] with text specific to the subagent. Do NOT MODIFY any other parts.
- Copy EXACTLY the parts other than [], including how to call the functions (e.g.
↪  "<function=example_function_here>")

Output format:
[Reasoning here...]
```yaml
system_template: |
  You are a helpful [role] assistant that can interact with a computer to [main task].
  <IMPORTANT>
  * If user provides a path, you should NOT assume it's relative to the current working directory.
  ↪  Instead, you should explore the file system to find the file before working on it.
  </IMPORTANT>

  You have access to the following functions:
  {{command_docs}}

  If you choose to call a function, you must ONLY reply in the following format with NO suffix:
  Provide any reasoning for the function call here.
  <function=example_function_name>
  <parameter=example_parameter_1>value_1</parameter>
  <parameter=example_parameter_2>
  This is the value for the second parameter
  that can span
  multiple lines
  </parameter>
  </function>
  (You must use the exact text function=" and "parameter=" for each function and argument, respectively,
  ↪  e.g. <parameter=command>value</parameter>)

  <IMPORTANT>
  Reminder:
  - Function calls MUST follow the specified format, start with <function= and end with </function>
  - Required parameters MUST be specified
  - CRITICAL: Only call ONE function at a time
  - Always provide reasoning for your function call in natural language BEFORE the function call (not
  ↪  after)
  </IMPORTANT>

  <pr_description>
  {{problem_statement}}
  </pr_description>

  CRITICAL: Use the submit_subagent function to provide the results when you are finished with your
  ↪  task.
  You are ONLY responsible for your specific assigned task. Do NOT attempt to resolve entire
  ↪  pr_description, only your task.
  Your goal is to complete your task in the MINIMAL NUMBER of steps. Resolve the issue fast and call
  ↪  submit_subagent as soon as possible.

instance_template: |-

  Your task:
  [Provide detailed, step-by-step instructions for your assigned subagent task, tailored to your
  ↪  specific role. The instructions must ONLY reference this subagent's function.]
  [If a context argument is provided, you MUST include its contents by inserting "{{context}}" here and
  ↪  explaining what the parameter is.]

  **CRITICAL: STAY IN YOUR LANE**
  - You are ONLY responsible for your specific assigned task
  - You are NOT responsible for solving the entire issue
  - You are NOT responsible for other subagent tasks
  - Focus EXCLUSIVELY on your assigned task and nothing else
  - CRITICAL: Call EXACTLY one function in your output!
  - CRITICAL: When you are finished, immediately call submit_subagent. Do not call any other tools or
  ↪  produce additional output.

  Focus exclusively on your assigned task and strictly follow these instructions. Do not attempt to
  ↪  address unrelated parts of the PR or perform work outside your specific subagent role.
  Use the submit_subagent tool after you are finished with your specific task to provide a clear and
  ↪  complete summary of your findings or changes.
  Your thinking should be thorough and so it's fine if it's very long.
```

Rules for generating templates:
1. The system_template should clearly define the subagent's role and capabilities based on the
↪  available tools
2. The instance_template should provide clear instructions for each task
3. Both templates should maintain consistent formatting with the base template
4. Ensure the templates encourage thorough analysis and clear documentation
5. MUST use literal block syntax: system_template: | and instance_template: |-
6. Never use quoted strings for templates
7. You should copy the given system template exactly other than the first sentence.
8. Modify the system template in the spots with [].

{{PREVIOUS_ITERATION_FEEDBACK}}
```

### A.1.3 CUSTOM ORCHESTRATOR PLAN PROMPT

**Prompt template for generating a custom orchestrator plan given a set of subagents**

```
You are a master workflow architect for automated software engineering. Your job is to design
↪  innovative, strategic workflows that maximize the effectiveness of available tools.

CONTEXT: You're designing workflows for an AI assistant that solves coding problems in software
↪  repositories. The assistant receives a problem description (like a bug report, feature request, or
↪  code issue) and needs to systematically work through the codebase to understand, fix, and validate
↪  the solution. The assistant has access to specialized "subagents" – each designed for specific
↪  aspects of the coding workflow.

You will be given a toolkit of specialized "subagents" – each with unique capabilities. Your challenge
↪  is to:
1. **Design** a comprehensive problem-solving plan that addresses the coding issue systematically
2. **Integrate** subagents strategically where they add the most value to your workflow
3. **Optimize** the sequence and wording of the plan to minimize the number of steps that the AI
↪  assistant takes while remaining effective

Think like a senior engineer designing a solution strategy. Consider:
- What are the key phases needed to solve this type of coding problem?
- Which subagents would be most valuable for specific phases?
- How can you combine subagent work with direct problem-solving phases?
- What's the most logical progression to integrate subagent input and output to solve the issue?
- How can you utilize subagents to minimize language model token usage and number of steps?

INPUT
- The available subagents will be provided inline between the following tags:
  <available_subagents>
  {{subagents_overview}}
  </available_subagents>
  The content in <available_subagents> lists each subagent with its name and short docs
  ↪  (summary/description). Treat it as the authoritative source for tool names and purposes.

WHAT TO OUTPUT
- Output ONLY your strategic plan as plain text (no YAML, no code fences, no headers).
- Each phase MUST start with a number and a period, e.g. "1. ...".
- For subagent phases, use the exact form: "Use the <name> subagent to ..."
- For direct phases, describe the action clearly without mentioning subagents, ensuring that it can be
↪  applied to any problem.
- Be creative and strategic – design workflows that combine different approaches effectively
- Keep 3 to 7 steps total, but make each step purposeful and well-reasoned
- Make sure the last steps are:
  - After you have solved the issue, delete any test files or temporary files you created.
  - Use the submit tool to submit the changes to the repository.
- Do not mention any function-call formats or system details

EXAMPLE (illustrative only; adapt to the given input)
<available_subagents>
- name: issue_localizer | Identify files and code regions relevant to the issue.
- name: error_reproducer | Reproduce the failing behavior and capture commands/outputs.
- name: code_tester | Run tests/commands to verify the fix and regressions.
</available_subagents>

Expected output EXAMPLE (plain text only):
1. Use the issue_localizer subagent to map the problem space and identify all potentially affected
↪  files and code regions.
2. Analyze the problem description and examine the identified files to understand the root cause and
↪  requirements.
3. Use the error_reproducer subagent to create a reproducible test case and capture the exact failure
↪  conditions.
4. Design and implement the fix based on the analysis, focusing on the specific files and code areas
↪  identified.
5. Use the code_tester subagent to validate the fix against the original failure case and run
↪  regression tests.
6. After you have solved the issue, delete any test files or temporary files you created.
7. Use the submit tool to submit the changes to the repository.

Now, based on the provided subagents, produce ONLY the numbered plan as plain text:
```

### A.1.4 PROMPT TEMPLATE FOR CHECKING IF A SUB-AGENT WAS HELPFUL IN A GIVEN INSTANCE

**Prompt template for checking if a subagent was helpful in a given instance**

```
CONTEXT:
These trajectories show a software engineering agent trying to fix a bug or implement a
↪  feature. The agent can use various tools including subagents (specialized AI
↪  assistants) to help solve the problem.

TRAJECTORIES:
{{TRAJECTORIES}}
```

```
TOOL TO ANALYZE: {{TOOL_NAME}}

Your task is to determine if the subagent "{{TOOL_NAME}}" was helpful in this set of
↪  trajectories.
A tool is considered helpful if:
1. It was called/invoked by the main agent in the main agent trajectory
2. It provided useful information, analysis, or insights that contributed to solving the
↪  problem
3. The main agent made progress after using this tool (e.g., identified the issue, made
↪  code changes, validated results, etc.)
4. It completed its task as intended (followed proper analysis process, not just got
↪  lucky results)

Look for positive evidence such as:
- The subagent being called with appropriate parameters
- The subagent providing insights that led to code changes or problem understanding
- The main agent referencing or building upon the subagent's output
- The subagent's output being used in subsequent reasoning or actions

Look for negative evidence such as:
- The subagent not being called by the main agent, or called incorrectly
- The subagent providing irrelevant or incorrect information that was not later used
- The subagent's response was valid but did not move the main agent closer to resolving
↪  the problem.
- The subagent failed to execute properly or had many errors during the its run.
- The subagent's output appeared correct, but its trajectory did not actually achieve
↪  those results (e.g., claimed to test code but just reported all tests passed).
- The main agent had to call the subagent over and over again to get the proper results.
- The subagent trajectory was unnecessarily long or verbose, taking many steps to
↪  complete its task
- The main agent trajectory became inefficient due to excessive subagent calls or overly
↪  verbose subagent responses
- The subagent's results did not actually help the main agent make progress in resolve
↪  the issue. If a subagent did not contribute to producing the correct patch, e.g. only
↪  improved performance, style, or documentation, this is NOT helpful.

Respond with YAML format (exactly):
```yaml
helpful: true/false
reasoning: |
  Brief explanation of why the tool was or wasn't helpful, including specific evidence
  ↪  from the trajectories
```

- Always use the block scalar `|` for `reasoning` and indent its text by two spaces.
- Only respond with the YAML block; no additional text before or after.
```

### A.1.5 ORCHESTRATOR-ONLY PROMPT

**Prompt template for generating an orchestrator prompt under orchestrator-only settings**

```
You are a master workflow architect for automated software engineering. Your job is to design effective
↪  workflows that solve coding problems efficiently.

CONTEXT: You're designing workflows for an AI assistant that solves coding problems in software
↪  repositories. The assistant receives a problem description (like a bug report, feature request, or
↪  code issue) and needs to work through the codebase to understand, implement changes, and validate
↪  the solution.

Your goal is to create workflows that are both effective at solving problems and efficient in their
↪  execution. Focus on designing strategic approaches that lead to successful problem resolution.

CONTEXT FOR PLAN USAGE:
Your generated plan will be inserted into this agent template context:

```
I've uploaded a python code repository in the directory {{working_dir}}. Consider the following PR
↪  description:

<pr_description>
{{problem_statement}}
</pr_description>
```

```
Can you help me implement the necessary changes to the repository so that the requirements specified in
↪   the <pr_description> are met? I've already taken care of all changes to any of the test files
↪   described in the <pr_description>. This means you DON'T have to modify the testing logic or any of
↪   the tests in any way! Your task is to make the minimal changes to non-test files in the
↪   {{working_dir}} directory to ensure the <pr_description> is satisfied. When solving the task,
↪   **first create a plan by breaking the problem into subtasks**. Think systematically about the steps
↪   needed to understand the problem, locate relevant code, implement changes, and verify the solution.
↪   Follow this process:
{{plan}}  <-- YOUR PLAN GOES HERE
You MUST follow the plan exactly.
```

AVAILABLE TOOLS:
- bash: Execute shell commands for searching, testing, running scripts, exploring codebase structure
- str_replace_editor: View, create, and edit files with precise string replacement capabilities
- submit: Submit the final solution

LEARNING FROM HISTORY:
<sampled_templates>
{{sampled_templates_summary}}
</sampled_templates>

If historical templates are provided above, identify what made the highest-scoring approaches
↪   successful and what caused failures. Look for patterns in tool usage, step efficiency, and
↪   problem-solving strategies. If no history exists, design breakthrough approaches that challenge
↪   conventional software engineering workflows.

WHAT TO OUTPUT:
- Create a step-by-step plan that an AI agent can execute systematically
- Each step must clearly specify tool usage ("Use bash to..." or "Use str_replace_editor to...") and
↪   expected outcomes
- Design for Python repositories and PR-based problem solving
- Focus on minimal, targeted changes rather than broad exploration
- Structure as numbered steps (1., 2., 3., etc.) with logical flow
- Final step must use submit tool to deliver the solution
- Make each step actionable and specific enough for precise execution
- Output ONLY the numbered plan as plain text (no formatting, headers, or explanations)
```

## A.2 SUB-AGENTS

### A.2.1 BOAD TOP 2 DISCOVERED SUB-AGENT CONFIGURATIONS FOR SEED-OSS-36B-INSTRUCT

---

**issue_analyzer configuration**

**docstring:**
[subagent] Analyzes and structures issue descriptions, bug reports, or feature requests to extract key
↪   information for systematic resolution planning. Best used at the beginning of issue resolution to
↪   understand requirements, extract technical details, and plan investigation approach. Outputs
↪   structured analysis including problem summary, affected components, reproduction criteria, success
↪   conditions, and recommended investigation approach. Repository state unchanged – only analyzes the
↪   provided issue context.

**argument:**
context (string) [required] : The complete issue description, bug report, feature request, or problem
↪   statement text that needs to be analyzed and structured. Include the full original text to ensure
↪   all requirements, technical details, constraints, and implementation notes are captured for
↪   systematic analysis.

**instance template:**
Your task: Analyze and structure the provided issue description to extract key information for
↪   systematic resolution planning, based on the provided context: {{context}}

Follow these steps:
1. Parse the issue description to extract core information: – Issue type (bug report, feature request,
↪   enhancement, etc.) – Problem statement and symptoms – Expected vs actual behavior – Error messages
↪   or failure descriptions – Affected functionality or components
2. Identify technical details: – File paths, function names, or code references mentioned – Stack
↪   traces or error logs – Version information or environment details – Dependencies or configuration
↪   issues
3. Structure reproduction criteria: – Steps to reproduce the issue – Required conditions or setup –
↪   Input data or test cases needed – Expected failure modes or symptoms
4. Define success conditions: – Clear criteria for issue resolution – Expected behavior after fix –
↪   Verification methods or tests needed – Performance or quality requirements5. Recommend
↪   investigation approach: – Priority areas to examine first – Suggested debugging or analysis methods
↪   – Related components that may be affected – Potential root cause categories.
6. Provide structured analysis including: – Concise problem summary – Categorized technical details –
↪   Clear reproduction steps – Measurable success criteria – Systematic investigation plan – Risk
↪   assessment and impact analysis
**CRITICAL: STAY IN YOUR LANE**– You are ONLY responsible for your specific assigned task– You are NOT
↪   responsible for solving the entire issue– You are NOT responsible for other subagent tasks– Focus
↪   EXCLUSIVELY on your assigned task and nothing else–
CRITICAL: Call EXACTLY one function in your output!–

---

```
CRITICAL: When you are finished, immediately call submit_subagent. Do not call any other tools or
↪   produce additional output.Focus exclusively on your assigned task and strictly follow these
↪   instructions. Do not attempt to address unrelated parts of the PR or perform work outside your
↪   specific subagent role.
Use the submit_subagent tool after you are finished with your specific task to provide a clear and
↪   complete summary of your findings or changes.
Your thinking should be thorough and so it's fine if it's very long.
```

### code_navigator configuration

```
docstring:
[subagent] Explores and maps relevant code structure in large repositories. Outputs structured
↪   information about key files, functions, classes, and their relationships. Repository state
↪   unchanged – only reads and analyzes code without modifications.

argument:
context (string) [required] : A string containing the issue description, error messages, stack traces,
↪   or specific code elements to investigate. Should include any relevant file paths, function names,
↪   class names, or keywords that might help locate the problematic code.

instance template:
Your task: Explore and map the relevant code structure in the repository based on the provided context:
↪   {{context}}
Follow these steps:
1. Parse the provided context to identify key elements to investigate (file paths, function names,
↪   class names, error messages, etc.)
2. Navigate through the repository structure to locate relevant files and directories.
3. Examine and analyze the identified code elements including: – Key files and their purposes –
↪   Important functions and their signatures – Classes and their methods/attributes – Module
↪   dependencies and imports – Code relationships and call hierarchies
4. Map the structure and relationships between different code components
5. Provide structured information including: – File locations and their roles in the codebase –
↪   Function/class definitions and their responsibilities – Dependencies between modules/components –
↪   Code patterns and architectural insights – Relevant code snippets that relate to the context
**CRITICAL: When you finish your analysis, immediately call submit_subagent with a comprehensive
↪   summary of your findings.**
**CRITICAL: STAY IN YOUR LANE** – You are ONLY responsible for your specific assigned task – You are
↪   NOT responsible for solving the entire issue – You are NOT responsible for other subagent tasks –
↪   Focus EXCLUSIVELY on your assigned task and nothing else – Do not attempt to address unrelated
↪   parts of the PR or perform work outside your specific subagent role
Use submit_subagent to provide a clear and complete summary of your findings when finished.
```

### A.2.2  ALL BOAD DISCOVERED SUB-AGENTS

Here we report all sub-agents discovered over 100 iterations of BOAD. For each sub-agent, we report the iteration number where it was first proposed (Generated Iteration), the number of times it was selected during optimization ($n$), and its final hindsight helpfulness rate.

| Sub-agent | Generated Iteration | $n$ | Helpfulness |
|---|---|---|---|
| code_navigator | 1 | 1140 | 0.933 |
| test_runner | 1 | 120 | 0.625 |
| code_fixer | 1 | 36 | 0.361 |
| fix_validator | 2 | 36 | 0.333 |
| issue_reproducer | 3 | 564 | 0.768 |
| dependency_resolver | 5 | 24 | 0.125 |
| test_analyzer | 6 | 24 | 0.083 |
| issue_analyzer | 7 | 1116 | 0.982 |
| precision_editor | 8 | 120 | 0.642 |
| multi_file_coordinator | 10 | 24 | 0.042 |
| code_detective | 11 | 60 | 0.500 |
| config_manager | 14 | 12 | 0.000 |
| git_resolver | 26 | 12 | 0.000 |
| error_debugger | 32 | 12 | 0.000 |
| api_analyzer | 36 | 24 | 0.208 |
| performance_analyzer | 46 | 36 | 0.250 |
| compatibility_checker | 48 | 36 | 0.333 |
| spec_analyzer | 53 | 36 | 0.361 |
| data_flow_analyzer | 60 | 96 | 0.562 |
| refactor_architect | 96 | 24 | 0.042 |

Table 4: Sub-agent statistics sorted by exp_num after 100 iterations of BOAD

### A.2.3 ALL EVOLUTION DISCOVERED SUB-AGENTS

Here we report all bundles discovered over 20 iterations of BOAD (one for each iteration). For each sub-agent, we report its iteration number, the generated sub-agents, and the bundle's success rate. Note that sub-agents with the same name across different iterations may have different system/instance templates because sub-agents are not reused. On evaluation, we choose the bundle with the highest success rate (latest bundle if tied).

Sub-agents focused on **problem analysis or file localization**—such as `issue_analyzer` (0.968 average helpfulness), `code_navigator` (0.917), and `issue_reproducer` (0.817)—consistently appear more useful. Our observation is that these agents provide value *independently* of how later stages unfold: even if code editing or testing fails, identifying the correct files and clarifying the underlying issue is almost always beneficial.

In contrast, sub-agents whose usefulness depends on earlier steps tend to show lower average helpfulness. For example, `test_analyzer` (0.167) is only useful *after* the system has already identified the faulty files and produced a candidate patch; if either prerequisite fails, it cannot contribute, which naturally lowers its average helpfulness.

Specialized sub-agents such as `dependency_resolver` (0.250) and `config_manager` (0.000) also have low average helpfulness, largely because dependency or configuration issues appear in only a small fraction of tasks.

Finally, once core analysis is completed and the faulty files are correctly located, the orchestrator can often complete the remaining code-editing steps on its own. This explains why code-editing sub-agents like `code_fixer` do not exhibit high average helpfulness.

| Iteration | Sub-agents | Success Rate |
|---|---|---|
| 1 | issue_localizer, issue_reproducer, patch_editor | 0.333 |
| 2 | fix_validator, fix_planner, code_explorer | 0.500 |
| 3 | issue_localizer, test_runner, code_patcher | 0.417 |
| 4 | repo_mapper, code_searcher, fix_validator | 0.500 |
| 5 | code_editor, issue_reproducer, issue_localizer | 0.182 |
| 6 | fix_validator, code_locator, problem_analyzer | 0.583 |
| 7 | test_runner, code_editor, code_analyzer | 0.583 |
| 8 | code_locator, fix_validator, issue_reproducer | 0.417 |
| 9 | code_editor, repo_analyzer, test_runner | 0.250 |
| 10 | code_locator, issue_reproducer, solution_planner | 0.417 |
| 11 | code_patcher, test_validator, repo_explorer | 0.417 |
| 12 | issue_reproducer, code_locator, fix_implementer | 0.364 |
| 13 | fix_validator, code_analyzer, code_editor | 0.500 |
| 14 | issue_reproducer, code_locator, bug_analyzer | 0.417 |
| 15 | test_validator, patch_editor, repo_explorer | 0.500 |
| 16 | code_locator, issue_reproducer, bug_analyzer | 0.455 |
| **17** | **test_validator, patch_editor, codebase_explorer** | **0.583** |
| 18 | issue_reproducer, bug_localizer, bug_analyzer | 0.250 |
| 19 | code_patcher, fix_validator, issue_analyzer | 0.417 |
| 20 | code_locator, reproducer, code_editor | 0.417 |

Table 5: Sub-agents and success rates per iteration for the evolutionary baseline.

### A.2.4 MANUALLY DESIGNED SUB-AGENT CONFIGURATIONS

---
**issue_localizer configuration**

**docstring:** A subagent that localizes the issue in the repository. Takes a context string specifying the
↪  brief description of the issue. Outputs a brief report about which files and lines are relevant to
↪  the issue.

**argument:** context (string) [required] { A string containing the brief description of the issue.

**instance template:**
Issue description:
{{context}}

---

```
Please identify which files and specific lines or functions are most relevant to this issue. Output a
↪   short, clear report that mentions:
- File paths
- Line numbers or function names
- A one-sentence explanation for why each location is relevant

Keep the report concise and focused on helping later agents work on the correct parts of the
↪   repository.
```

## error_reproducer configuration

**docstring**: A subagent that creates and executes test scripts to verify reported errors. Outputs the
↪   result of the tests and locations of test files created.

**argument**: context (string) [required] { A string containing error details, file paths and line numbers
↪   of code relevant to the error, and expected vs actual behavior.

**instance template**:
```
Error context:
{{context}}

Please create and execute a temporary reproduction script to verify this error. You should:
- Create temporary files (prefixed with 'tmp_')
- Include only what's needed to reproduce the error
- Report whether the error reproduces exactly as described
- Note any deviations from expected behavior

Output a short, clear report that mentions:
- Result of the tests
- Locations of test files created
```

## code_editor configuration

**docstring**: A subagent that implements specified code changes in the repository. Outputs the specific
↪   code changes made, file paths/line numbers edited, and what should be tested to verify the fix.

**argument**: context (string) [required] { A string containing the code snippet(s) to modify and file
↪   path(s), and the changes to be applied.
**instance template**:
```
Context for code changes:
{{context}}

Please implement the specified code changes in the repository. You should:

- Identify the relevant files and code sections
- Make precise edits according to the specification
- Maintain code quality and consistency
- Output a short, clear report that mentions:
- File paths/line numbers edited
- What should be tested to verify the fix
```

## code_tester configuration

**docstring**: A subagent that tests code after edits have been made to verify the fix works correctly.
↪   Outputs the result of the tests.

**argument**: context (string) [required] { A string containing the specific code changes made, file paths,
↪   and original error.

**instance template**:
```
Code changes made:
{{context}}

Please test the code after the edits to verify the fix works correctly. You should:

Use existing test files if available (prefixed with 'tmp_'), or create new ones as needed

- Test the specific functionality that was changed
- Determine whether or not the original error is fixed.
- Output a short, clear report that mentions:
- List of tests run and results of the tests
- Whether the original error is fixed, and if any new errors were introduced
```

## A.3 ADDITIONAL EXPERIMENTAL RESULTS

### A.3.1 EFFECT OF DESIGN-SET SIZE

We evaluated whether the size of the design set affects the performance of the discovered sub-agents and found no meaningful impact. We sample 6 unique problems from SWE-Bench-Verified and sample one problem from each to make a small design set of 6 problems, and sample two problems from each to make a large design set of 24 problems. Across different design-set sizes, the resulting sub-agents achieve similar performance as shown in Table 6.

| Design Set Size | Resolution Rate |
|---|---|
| 6 | 21% |
| 12 | 20% |
| 24 | 19% |

Table 6: Performance of discovered sub-agents across different design-set sizes.

## A.4 ADDITIONAL EXPERIMENTAL SETUP DETAILS

### A.4.1 IMPLEMENTATION DETAILS FOR EVOLUTIONARY BASELINE

For comparison with BOAD, we implemented an evolutionary multi-agent design baseline where the orchestrator and three sub-agent prompts are bundled together and treated as a single evolutionary individual. We use three sub-agents paired with an orchestrator because BOAD chooses three sub-agents when optimizing the sub-agents in the experiments (Section 5.2). At each iteration of the evolution, the system generates three new sub-agents sequentially, given the previous round sub-agent configurations and their measured helpfulness and success rates (using the same prompts as BOAD, provided in A.1.2). Next, both the sub-agent warm-up and the orchestrator construction follow the same procedures used in BOAD. The generated orchestrator and sub-agents are then run on the same design set to get the helpfulness and success rate for the next iteration.

### A.4.2 API COST COMPARISON WITH BOAD

The evolutionary baseline also requires Claude calls—both to propose new orchestrator/sub-agent prompts and to perform LLM-as-judge scoring. Under the same setup, each evolution iteration costs $2.33, whereas each BOAD iteration costs $0.96, indicating that evolution is substantially more expensive to run. Evolution incurs higher cost because it mutates the bundle of orchestrator and sub-agents at each iteration.

## B THE USAGE OF LARGE LANGUAGE MODELS (LLMS)

In our work, we used large language models (LLMs) for two purposes: (1) as a general writing assistant to check the grammar of the manuscript for readability, and (2) as the model component of our proposed system BOAD, where LLMs function as the evolution engine, meta-/sub-agents, and evaluation judges in experiments with mainstream coding agents. All research ideas, methodological contributions, and conclusions are solely those of the authors, who take full responsibility for the content of this work.

