# OpenReview forum: "BOAD: Discovering Hierarchical Software Engineering Agents via Bandit Optimization"
_ICLR.cc/2026/Conference — ICLR 2026 Poster_

### Official Review · Reviewer_xwJx · 2025-10-30

**Soundness:** 3
**Presentation:** 3
**Contribution:** 3
**Rating:** 8
**Confidence:** 4

**Summary:**

The paper addresses the problem of automated development of SWE-Agents. They consider a multi-agent setup structured such that an orchestrator agent calls multiple sub-agents. The main concern of the paper is optimizing the start-up prompts for the orchestrator and subagents. They treat this as a bandit problem, maintaining and growing a set of sub-agents over time. Each sub-agent is scored using a with an LLM credit assignment approach. The paper's evaluations show strong results on SWE-Bench Verified and Live.

**Strengths:**

The major strength of the work is the novel formulation of the problem as an MAB problem and the demonstration different aspects of the  formulation leads to improvement of performance.

1. The framing of optimising multi-agent systems as multi-armed bandits is novel in the context of SWE-Agents. The work successfully formulates a UCB strategy for this context as well as a way to grow the number of subagents. Ablations demonstrate how optimising sub-agents explicitly outperforms static approach.

2. The use of LLM based credit assignment technique in the multi-armed bandit setup is novel and very interesting here. It shows a way to learn from specific mistakes the system might be making in a modular way. Ablations also support the importance of this metric.

3. Evaluation show improvements on standard benchmarks, especially on SWE-Bench Live.

**Weaknesses:**

1. The major limitation of this work is that it is limited to only improving the starting prompt of the system. From the difference in performance of scaffolds present on https://www.swebench.com/, it is reasonable to believe that design decisions can impact the behavior of agent, as much or maybe more than prompts.

2. The analysis of the behaviour of the system is limited to short qualitative analysis. This is currently not substantiated with example or quantitative metrics. There is also no exposition of what sorts of agents are "discovered".

**Questions:**

1. Related to W1, can this idea be extended to agent design as well as prompt content?

2. Related to W2:
a) Is there quantitative evidence for multi-agent systems being better in the sorts of ways described in 5.4
b) What sorts of agents are "discovered" and is there a way to characterise the trajectory of the system as is discovers these?

---

> ### Author Response · Authors · 2025-11-21
>
> We're glad you appreciate the novelty of formulating multi-agent design as an MAB problem and applying it to the SWE context! Thank you for acknowledging the strong empirical performance of our work. We address your remaining questions below.
>
> > The major limitation of this work is that it is limited to only improving the starting prompt of the system. From the difference in performance of scaffolds present on https://www.swebench.com/, it is reasonable to believe that design decisions can impact the behavior of agent, as much or maybe more than prompts.
> >
>
> > Related to W1, can this idea be extended to agent design as well as prompt content?
> >
>
> We appreciate this insightful observation! You're absolutely right that scaffolding can be even more impactful than prompts alone. The good news is that our algorithm's framing is naturally extensible to scaffolds as well—it's agnostic to what's being optimized, whether prompts, scaffolds, or other design elements. We're excited about exploring this direction in future work, where we plan to expand the design space to include scaffolding itself, potentially unlocking even greater performance improvements.
>
> > The analysis of the behaviour of the system is limited to short qualitative analysis. This is currently not substantiated with example or quantitative metrics. There is also no exposition of what sorts of agents are "discovered".
> >
>
> > Related to W2:
> >
>
> > a) Is there quantitative evidence for multi-agent systems being better in the sorts of ways described in 5.4
> >
>
> Thank you for raising this excellent point! To quantitatively support our observations in Section 5.4, we analyze patch length statistics on SWE-Bench-Live (shown below). The results are striking: BOAD's discovered multi-agent system produces substantially shorter and more localized patches than the single-agent baseline. Specifically, the average patch length for BOAD is less than 1/3 of the baseline's length. Most notably, this difference stems from adding significantly fewer lines (197 vs. 855)—a direct reflection of the over-editing behavior we described.
>
> Patch length statistics on SWE-Bench-Live runs:
>
> |  | BOAD-discovered Multi-agent | Single-agent (SWE-agent) |
> | --- | --- | --- |
> | total lines (mean) | 258 | 931 |
> | total lines (median) | 120 | 435 |
> | total characters (mean) | 10264 | 32949 |
> | total characters (median) | 5157 | 15782 |
> | added lines (mean) | 197 | 855 |
> | removed lines (mean) | 11 | 15 |
>
> > b) What sorts of agents are "discovered" and is there a way to characterise the trajectory of the system as is discovers these?
> >
>
> We have added the full set of discovered sub-agents to Appendix A.2.1 in the order they were generated. Qualitatively, BOAD produces broader sub-agents early in the search (e.g., code_fixer, issue_reproducer). Later iterations discover new sub-agent roles (e.g., issue_analyzer, which analyzes issues that emerged after 7 iterations) or introduce more specialized and less obvious tools (e.g., data_flow_analyzer, multi_file_coordinator, compatibility_detector).

---

> > ### Author Response · Authors · 2025-11-25
> >
> > We hope the clarifications above address the reviewer’s questions. We’re more than happy to continue the conversation and provide any additional details that would be helpful.

---

### Official Review · Reviewer_rVvm · 2025-10-31

**Soundness:** 3
**Presentation:** 4
**Contribution:** 3
**Rating:** 6
**Confidence:** 3

**Summary:**

This paper searches for prompts of sub-agents for software engineering. The resulting agent consists of a set of sub-agents and an orchestrator (all parameterized by prompts). During inference, the agent will ask the orchestrator to call sub-agents sequentially to solve a problem. To search for the set of sub-agents, it maintains a pool of candidate sub-agents, picks and evaluates a subset of candidates each time, and updates the sub-agents' information according to the evaluation results afterward. The paper uses UCB to balance exploration and exploitation during search. It can also add a new sub-agent each round to improve diversity. LLM-judge is used to assign credits to each sub-agent as well. The searched agent is evaluated on two popular benchmarks and demonstrated strong performance.

**Strengths:**

The method is intuitive and interesting, although it largely relies on the abilities of LLMs for judging and proposing sub-agents. We do need various ways to balance exploration and exploitation. The searched agent demonstrates strong performance on popular benchmarks as well.

The paper is generally well-written and easy to understand.

**Weaknesses:**

* Missing naive baseline, such as evolution search. One can treat all prompts of sub-agents and/or orchestrators as parameters and use LLMs + evolution search to optimize them. There are various prior works that balance exploration and exploitation for naive LLM tree search as well. The authors discussed this baseline in the method section, claiming it is prohibitively expensive with no experimental results.
* I'm not sure if the comparison with baselines is fair, missing experimental details. It would be great to include the costs of calling Claude-4 (which model?) for each method. I am not sure if the method is stochastic or deterministic either. If not deterministic (hard to imagine LLM to be deterministic even with temperature=0), how noisy is the evaluation?
* The meta-prompts (e.g., for proposing and refining sub-agents) look long and detailed. Can the method discover novel sub-agents that are not expected, given the meta-prompts?
* It would be great to include the searched prompts for various methods as well.

**Questions:**

* Are there results of the naive evolution search?
* Can you provide more experimental details such as the variance of evaluation, the cost for each method, and meta-prompts for each method?
* Can you share the final best-performance prompts of the searched sub-agents?

---

> ### Author Response · Authors · 2025-11-21
>
> Thank you for the thoughtful and positive feedback. We’re glad the paper was easy to follow and that you found the method interesting. We appreciate that the reviewer acknowledges the strong performance of our method. We address the remaining questions below.
>
> > Evolution search baseline
> >
>
> We appreciate the reviewer's suggestion to include a comparison with an evolutionary baseline. To address this, we implemented an approach adapted from ADAS ([22] in the manuscript), where the orchestrator and sub-agent prompts are evolved jointly.
>
> Both BOAD and the evolutionary baseline were run under the same evaluation budget, meaning each method was allowed the same number of design-set evaluations in the SWE-Bench environment. Each evaluation runs the full multi-agent system on a problem and produces a patch. The evolutionary baseline also incurs higher Claude API usage during search, as it generates a new bundle of orchestrator and sub-agent prompts at every iteration. The evolutionary baseline reaches a 17% resolution rate on SWE-Bench-Live, whereas BOAD reaches 20%. This indicates that BOAD makes more effective use of the available search budget and API calls. Notably, on the SWE-Bench-Live leaderboard, only one agent exceeds 20%, while many achieve around 17%, demonstrating that gains at this performance level are particularly challenging. We report this comparison in Table 1 and provide implementation and cost details in Appendix A.4.
>
> > Experimental details and Claude API cost
> >
>
> Thank you for raising this important point. To clarify: Claude models are used only during the search phase for optimizing sub-agents on a small design set, never at test time. The test-time comparison in Table 1 is fully controlled and fair—our method and both baselines (the single-agent SWE-agent and the manually designed sub-agent baseline) all use the same underlying model, Seed-OSS-Instruct. The only difference lies in how the orchestrator and sub-agent prompts are obtained, and whether sub-agents are used at all.
>
> We additionally report results from larger models (e.g., Claude + SWE-agent) to highlight that BOAD enables smaller models to match or even outperform much larger systems. We have clarified this motivation in the revision.
>
> Regarding cost, the evolutionary baseline introduced during rebuttal also relies on Claude to propose new orchestrator/sub-agent prompts and to perform LLM-as-judge scoring. Under the same setup, each evolution iteration costs $2.33, compared to $0.96 for BOAD. In practice, evolution also required more wall-clock time, because it proposes many new candidates and evaluates them by running full test cases.
>
> > How noisy is the evaluation?
> >
>
> We appreciate your question about evaluation consistency due to the non-determinism of LLM, even with temperature 0. To verify the stability of our results, we ran the same multi-agent design (orchestrator + sub-agents) on SWE-Bench-live three times. The results showed good consistency, with a mean of 0.193 and a standard deviation of  0.0097, closely matching the 20% resolution rate reported in Table 1.
>
> > Can the method discover novel sub-agents that are not expected, given the meta-prompts?
> >
>
> Yes, the LLM can generate sub-agents beyond the example types shown in the meta-prompt (Appendix A.1.2). While the meta-prompt provides only a few example steps of the workflow (localization, issue reproduction, testing), BOAD discovers a significantly more diverse set of behaviors. For example, BOAD discovered `dependency_resolver` in its 5th iteration—a sub-agent that identifies dependency-related issues from the issue description. Other novel sub-agents include `multi_file_coordinator` and `performance_analyzer`. These types of sub-agents were never explicitly requested in the meta-prompt. We have added a list of all BOAD discovered sub-agents to the manuscript (Appendix A.2.2).
>
> > It would be great to include the searched prompts for various methods as well.
> >
>
> We have added all the sub-agents discovered by our BOAD method in A.2.2 and the evolution baseline in Appendix A.2.3 as well as those used in the "Manual" sub-agent baselines (from Table 1) in Appendix A.2.4.

---

> > ### Author Response · Authors · 2025-11-21
> >
> > > Can you provide more experimental details such as the variance of evaluation, the cost for each method, and meta-prompts for each method?
> > >
> > - Variance of evaluation: We have included the variance results above in our response.
> > - Cost of each method: For methods where we use results directly from the SWE-Bench-Verified and SWE-Bench-Live leaderboards, we do not have access to their execution trajectories and therefore cannot compute token usage or API cost. For the baselines we ran ourselves, we report their token usage in Table 2.
> > - Meta-prompts: For BOAD and the evolutionary-search baseline, the meta-prompts used are the same. We provide these in Appendix A.1.2. The baseline, which uses manually designed sub-agents, does not employ meta-prompts, as we directly adapt the sub-agent prompts and roles from prior work.
> >
> > > Final best-performance prompts of the searched sub-agents
> > >
> >
> > We’ve added the best-performing sub-agents’ prompts in Appendix A.2.2.
> >
> > We appreciate these suggestions, and we have incorporated all the requested details into the revised manuscript.

---

> > > ### Author Response · Authors · 2025-11-25
> > >
> > > We hope the clarifications above address the reviewer’s questions. We’re more than happy to continue the conversation and provide any additional details that would be helpful.

---

### Official Review · Reviewer_Mz7z · 2025-11-01

**Soundness:** 2
**Presentation:** 2
**Contribution:** 2
**Rating:** 6
**Confidence:** 4

**Summary:**

This paper proposes BOAD, a method for discovering and selecting subagents to use as tools for an orchestrator agent in software engineering tasks. It considers subagents in a set as arms of a multi-armed bandit and assign credits with both the test-based final outcome reward and LLM-as-a-judge process reward. After running this bandit optimization on 12 issues from SWE-Bench Verified, the discovered multi-agent system can perform well on both SWE-Bench Verified, and SWE-Bench Live, which doe not have repository-level overlap with the problems in SWE-Bench Verified.

**Strengths:**

1. Interesting formulation of the multi-agent system discovery problem as a multi-armed bandit. This creates balance between exploration and exploitation while selecting and evolving subagents.

2. Great performance on SWE-Bench Live demonstrates the effectiveness of the method.

3. Comprehensive ablation showing the effectiveness of having the subagents, customizing the orchestrator, and using hindsight helpfulness for credit assignment.

**Weaknesses:**

1. Lack of details about the actual optimization process. How many agents are there in the final set of subagents? How many of the top agents are from the expanded set or the initial set of subagents? What are the final top agents selected? Qualitatively why are they better than other subagents? Readers need these details to get a better idea of the final discovered system.

2. Single run evaluation is insufficient. The non-determinism of LLM agents results on lots of randomness in every agent run, which impacts the optimization process. Is BOAD always effective as reported or will its performance fluctuate across multiple runs?

**Questions:**

1. Any plans to compare against manual optimization conducted by human engineers in the loop?

---

> ### Author Response · Authors · 2025-11-21
>
> Thank you for the insightful and encouraging feedback. We’re delighted that the bandit-based formulation of multi-agent discovery resonated with you, and we truly appreciate your recognition of the strong SWE-Bench Live performance and the depth of our ablation studies. We clarify the remaining questions below.
>
> > How many agents are there in the final set of subagents?
> >
>
> We found that selecting the top two sub-agents ranked by helpfulness yields the best performance. Table 3 summarizes the results across different numbers of sub-agents.
>
> > How many of the top agents are from the expanded set or the initial set of subagents?
> >
>
> Among the top five sub-agents, two come from the initial set and three come from the expanded set. The top two sub-agents we ultimately selected include one discovered in the initial iteration and one discovered in the 7th iteration. We added the list of all discovered subagents in Appendix A.2.1.
>
> > What are the final top agents selected?
> >
>
> We select sub-agents based on the average helpfulness score (Section 4.2). The top two sub-agents are `issue_analyzer` and `code_navigator`. The `issue_analyzer` reasons about the root cause of the issue, while `code_navigator` locates faulty files. We added details on the selection process to the updated manuscript (at the end of Section 5.1) and included the full prompts for the top two sub-agents in Appendix A.2.2.
>
> > Qualitatively why are they better than other subagents?
> >
>
> Sub-agents focused on **problem analysis or file localization**—such as `issue_analyzer` (0.968 average helpfulness), `code_navigator` (0.917), and `issue_reproducer` (0.817)—consistently appear more useful. Our observation is that these agents provide value *independently* of how later stages unfold. Even if code editing or testing fails, identifying the right files and clarifying the problem is almost always helpful.
>
> In contrast, sub-agents whose usefulness depends on earlier steps tend to show lower average helpfulness. For example, `test_analyzer` (0.167) is only useful *after* the system has already identified the faulty files and produced a candidate patch; if either prerequisite fails, it cannot contribute, which naturally lowers its average helpfulness.
>
> Specialized sub-agents such as `dependency_resolver` (0.250) and `config_manager` (0.000) also have low average helpfulness, but this is likely because dependency or configuration issues appear in only a small fraction of tasks.
>
> Finally, once the core analysis is done and the faulty files are correctly located, the orchestrator can often complete the remaining code-editing steps on its own. This explains why code-editing sub-agents like `code_fixer` do not exhibit high average helpfulness.
>
> We have added this analysis to Appendix A.2.3.
>
> > Single run evaluation is insufficient. The non-determinism of LLM agents results on lots of randomness in every agent run, which impacts the optimization process. Is BOAD always effective as reported or will its performance fluctuate across multiple runs?
> >
>
> Thank you for raising this point. We agree that single-run evaluations can be sensitive to LLM stochasticity, even at temperature 0. We are conducting additional runs to verify robustness.
>
> So far, we have completed one additional full BOAD run. The resulting multi-agent design achieves a **20%** success rate, consistent with the original result.
>
> Although additional runs on the original design set are still in progress, our design-set ablation (Appendix A.3.1) already demonstrates robustness to randomness. We varied the design-set size (6, 12, and 22 randomly sampled problems from SWE-Bench-Verified; the manuscript uses 12). Across all settings, the discovered agents achieve **19–21%** on SWE-Bench-Live. This consistency shows that BOAD's performance is robust to randomness and the inherent non-determinism of LLM agents.
>
> > Any plans to compare against manual optimization conducted by human engineers in the loop?
> >
>
> Thank you for the thoughtful suggestion. We absolutely agree that comparing against human-in-the-loop manual optimization would be valuable and could provide important insights. In this work, our focus has been on evaluating fully automated methods, but we very much appreciate this direction and would be excited to explore human-guided baselines in future research.
>
> We sincerely appreciate your thoughtful questions and your engagement with the details of our approach. Your feedback has helped us strengthen the clarity and completeness of the manuscript, and we’re grateful for your careful evaluation. We hope the revised version addresses your concerns fully.

---

> > ### Author Response · Authors · 2025-11-25
> >
> > We hope the clarifications above address the reviewer’s questions. We’re more than happy to continue the conversation and provide any additional details that would be helpful.

---

> ### Comment · Reviewer_Mz7z · 2025-11-26
>
> Thank you for the clarification. It makes things much clearer.
>
> One remaining question I have regarding BOAD is, does this seem like an overkill? The framework itself looks very general and applicable to lots of subagents, but at the end of process only two agents are picked and their functionality is not really surprising. Is it really necessary to use such an elaborate method to get to these two agents?

---

> > ### Author Response · Authors · 2025-11-27
> >
> > We’re very glad the earlier clarification was helpful! On the remaining question:
> >
> > > One remaining question I have regarding BOAD is, does this seem like an overkill? The framework itself looks very general and applicable to lots of subagents, but at the end of process only two agents are picked and their functionality is not really surprising. Is it really necessary to use such an elaborate method to get to these two agents?
> > >
> >
> > That’s a great question. When we let the LLM generate sub-agents, it produces a whole range of candidates that *all* look sensible at first glance—things like `test_analyzer`, `dependency_resolver`, `patch_refiner`, and multiple versions of `issue_analyzer` or `code_navigator`. Each comes with a perfectly reasonable description, and it’s not obvious which ones will meaningfully help the system.
> >
> > What we found is that many of these “plausibly helpful” agents don’t end up contributing much in practice. BOAD gives us a simple way to sort through these LLM-generated sub-agent designs and identify the ones that actually matter.
> >
> > So while the final two agents may look intuitive in hindsight, arriving at them from a large pool of reasonable candidates isn’t straightforward. BOAD makes that search tractable and reliable, turning a broad design space into a focused set of genuinely useful sub-agents.
> >
> > We hope this clarification is helpful, and we’d be glad to answer any further questions.

---

### Official Review · Reviewer_5qkT · 2025-11-03

**Soundness:** 3
**Presentation:** 3
**Contribution:** 3
**Rating:** 4
**Confidence:** 2

**Summary:**

Previous work in agentic frameworks for code generation have shown a worrying trend -- performance drops significantly from in-distribution benchmarks (SWE-Bench Verified) to out-of-distribution benchmarks (SWE-Bench-Live). The authors posit that one of the root causes for this is the long-horizon and complex nature of SWE-tasks, which makes it hard for single-agent frameworks to delegate and solve tasks. Furthermore, current multi-agent systems rely on expensive evolutionary optimization algorithms, which is impractical in the current domain.

The authors hypothesize that a multi-armed bandits (MAB) presents an efficient framework to build and optimize such multi-agent systems to solve such tasks. Specifically, they propose BOAD, which treats hierarchical multi-agent systems as a sequential decision-making process with the reward signal for evaluating each sub-agent (each arm in the MAB) coming from an LLM-as-a-judge.

BOAD achieves impressive performance on SWE-Bench Verified and SWE-Bench-Live.

**Strengths:**

- The paper is very well written and I thank the authors for explicitly stating the MAB formulation in the context of the code generation task.
- In terms of novelty, I was pretty surprised but a multi-arm bandit optimizer hasn't been tried before for software engineering tasks.

**Weaknesses:**

**Methodology:**
* **Training using 12 problems**: My understanding is that the SOTA accuracy number was reached after including 12 problems from SWE-Bench-Verified in the design set for BOAD. This raises two concerns:
	* Why 12 problems specifically? Does increasing or decreasing the set of design problems drastically effect final performance?
	* None of the other baselines are automatically tuning the multi-agent system. I understand that an evolutionary agent here might be less efficient than BOAD, but including a result for how BOAD fairs against an evolutionary agent would undoubtedly present a useful point of comparison to understand the tradeoff.

* **Efficiency Experiment:** One of the motivating points of using BOAD (from the introduction) is that it is hypothesized to be more efficient than evolutionary multi-agent frameworks at the same task. However, the Token Analysis experiment is not enough to justify the efficiency for two reasons:
	* There isn't a direct comparison against an evolutionary multi-agent framework.
	* Efficiency can be achieved by either reducing token usage OR by reducing the underlying model size:
		* Specifically, In terms of floating point operations, querying a smaller model more times is more efficient than querying a larger model less times.
		* For example (be advised: this is a rough analysis without access to underlying data):
			* Assuming equivalence in all other aspects, a `30B` model can generate `20%` more tokens than a similar `36B` model yet have the same total cost.
			* Under this logic, the OpenHands single-agent that uses `Qwen3-coder-30B` and achieves a 51.6% resolution rate for SWE-bench Verified in is actually *more efficient*  than SWE-Agent+BOAD using the `OSS-36B` model which achieves a 53.2% resolution rate.
		* My recommendation:
			* Getting access to TFLOP data is extremely hard. Instead, most works use normalized cost, which is defined as the fractional cost of querying a smaller model per token compared to the cost of querying the largest model. e.g. if cost for 36B model is `1` per token, then cost for a 30B model will be `0.8334`.
			* Then, such methods will show an efficiency-performance tradeoff curve.
			* The key insight is to demonstrate that the efficient-agent algorithm is at the pareto froont of the tradeoff curve for different configurations.
			* Look at Figure 3 of this paper [https://arxiv.org/abs/2504.07247](https://arxiv.org/abs/2504.07247) for an example of how this is computed and analyzed. This might be a good paper for the related works as well.


**Overall:** I'm slightly leaning towards rejecting the paper. The results are pretty impressive and I generally haven't seen a similar mutli-agent system optimized with bandits before, but there are some issues with the experimental setup that need to be better understood before proceeding. I'm happy to discuss these points with the authors during the discussion period.

**Questions:**

Look at weaknesses.

---

> ### Author Response · Authors · 2025-11-21
>
> We thank the reviewer for the thoughtful and encouraging feedback. We’re glad the motivation and ideas came through clearly. We’re also encouraged that the reviewer found the method intuitive and the results strong. Thank you for the constructive review. We will address the remaining questions below.
>
> ## Design set size
>
> > Why 12 problems specifically?
> >
>
> SWE-Bench-Verified is highly imbalanced, with a few repositories dominating the dataset. Since it contains 12 unique repositories, we selected one problem from each to create a balanced design set. These 12 problems do not overlap with SWE-Bench-Live. We report our method's resolution rate on SWE-Bench-Verified for both the held-out set (53.1%) and the full set (53.2%).
>
> > Does increasing or decreasing the set of design problems drastically effect final performance?
> >
>
> We evaluated whether the design set size affects the performance of discovered sub-agents and found no significant impact. We sampled 6 unique problems from SWE-Bench-Verified—one from each repository—to create a small design set. We also sampled two problems from each repository to create a large design set. Since two repositories contain only one problem each, the large design set totals 22 problems. Across all design set sizes, the resulting sub-agents achieve similar performance, as shown in the table below.
>
> | Design set size | Resolution Rate |
> | --- | --- |
> | 6 | 21% |
> | 12 | 20% |
> | 22 | 19% |
>
> ## Comparison with Evolutionary Baselines
>
> We appreciate the reviewer's suggestion to include a comparison with an evolutionary baseline. To address this, we implemented an approach adapted from ADAS ([22] in the manuscript), where the orchestrator and sub-agent prompts are evolved jointly.
>
> Both BOAD and the evolutionary baseline were run under the same evaluation budget, meaning each method was allowed the same number of design-set evaluations in the SWE-Bench environment. Each evaluation runs the full multi-agent system on a problem and produces a patch. The evolutionary baseline incurs higher Claude API usage during search, as it generates a new bundle of orchestrator and sub-agent prompts at every iteration. The evolutionary baseline reaches a 17% resolution rate on SWE-Bench-Live, whereas BOAD reaches 20%. This indicates that BOAD makes more effective use of the available search budget and API calls. Notably, on the SWE-Bench-Live leaderboard, only one agent exceeds 20%, while many achieve around 17%, demonstrating that gains at this performance level are particularly challenging. We report this comparison in Table 1 and provide implementation and cost details in Appendix A.4.
>
> ## Efficiency
>
> We appreciate the reviewer's thoughtful comments on efficiency. We apologize for the unclear wording. In the paper, we use "efficiency" to refer specifically to **search efficiency**—how effectively a method improves the multi-agent design under a fixed *search* budget.
>
> ### Search Efficiency
>
> During the search phase, BOAD and the evolutionary baseline use the same underlying model and the same number of evaluation calls on the same design set. Within this setting, BOAD arrives at a stronger final design (20% vs. 17%). In this setting, both methods use the same model, so the comparison naturally focuses on how each method uses the same evaluation budget. We will clarify this definition in the revised manuscript.
>
> The reviewer’s suggestion to consider a Pareto frontier is greatly appreciated. This viewpoint can be applied to search efficiency by examining the tradeoff between performance and the number of evaluation calls. We are computing these results and will include the corresponding curve in the updated version.
>
> ### Token Analysis Purpose
>
> The token analysis in Table 2 aims to demonstrate that the use of sub-agents reduces context length for each LLM call by decomposing tasks, and this effect stems from the multi-agent architecture rather than from BOAD specifically. Importantly, this efficiency differs from search efficiency  because it's measured at test-time when deploying the resulting system to solve specific SWE-Bench tasks. We’ve made this clearer in the revision.
>
> We thank the reviewer again for the constructive and thoughtful feedback. Your comments helped us clarify the presentation and refine our experiments, and we’re grateful for the opportunity to strengthen the paper. We hope the revisions address your questions and that the updated draft better reflects the contributions of our work.

---

> ### Comment · Reviewer_5qkT · 2025-11-21
>
> Thank you for the additional details!
>
> > design set = 12 unique repositories
>
> I see. The `12` number makes more sense now.
>
> > increasing design set from 6 -> 22 doesn't change performance much (actually, it decreases performance)
>
> Fascinating. What if the design set was even smaller (i.e. subset of the SWE-Bench Verified problems)? What's the lowest number of problems before we see a reduction of performance? Does the subset matter?
>
> > evolutionary agent
>
> Thank you for the additional details! The experiment setting makes sense. A larger comparison should be a great result in the paper (i.e: let the evolutionary agent converge, and then give BOAD the same steps and see if it does better), but that might be prohibitively expensive so this should be enough.
>
> > efficiency = search efficiency
>
> This is my mistake. Thank you for the clarification.
>
>
> Happy to increase score to 6.

---

> > ### Author Response · Authors · 2025-11-25
> >
> > We’re glad our earlier clarifications were helpful, and we truly appreciate the reviewer’s thoughtful suggestions. The design-set experiments you proposed have already given us deeper insight into our method, and we’re now extending them by testing additional design-set configurations. We are also running the evolutionary baseline for longer to observe its behavior at convergence and to compare it more thoroughly with BOAD. We will add them in the revision.

---

### Author Response · Authors · 2025-12-01

Dear Area Chair,

Thank you again for handling the process, especially in light of the recent OpenReview incident. We’re very grateful for the time and care you and the reviewers have invested in our paper.

Overall, the reviewers were excited about the core ideas: they highlighted the **novelty** of framing multi-agent design as a bandit problem, the use of LLM-based credit assignment, and the **strong empirical performance** on the challenging benchmark SWE-Bench-Live. Below is a short summary of how we addressed their remaining questions.

---

### Reviewer 5qkT (score 4, increased to 6)

The reviewer found the bandit formulation both novel and natural and described the results as “pretty impressive.” Their remaining questions are addressed below.

**Design set size (why 12 and sensitivity):** The reviewer asked why we chose this size and whether the method is sensitive to it; we explained that the design set was balanced across repositories and that systems discovered using smaller or larger sets achieve similar performance, suggesting the method is stable.

**Comparison with evolutionary baselines:** The reviewer asked for a comparison with an evolutionary search; we implemented one under the same search budget and found that our BOAD method consistently achieves stronger results while also requiring fewer API calls. We added these results to the manuscript.

**Meaning of “search efficiency”:** The reviewer asked what we meant by “efficiency,” and we clarified that it refers to how quickly performance improves during the search stage under the same search budget, which is different from the test-time token usage of the final system.

The reviewer noted that these clarifications resolved their concerns and raised their score accordingly.

---

### Reviewer Mz7z (score 6)

This reviewer was very positive about the formulation and the results, and their remaining questions were straightforward to address.

**Details of the final discovered system:**
They asked for a clearer picture of the final multi-agent system, and we explained that it ultimately consists of the two most consistently helpful sub-agents, with a short overview of how they emerged during the search. We also added the full prompts and a complete list of all discovered sub-agents to the appendix for completeness.

**Why are these agents better than others qualitatively:**
They were curious why certain sub-agents rise to the top, and we clarified that agents focused on understanding the issue and locating relevant files naturally contribute more often, whereas more situational agents appear less frequently. We included intuitive examples to make this pattern easy to follow.

**Stochasticity and single-run evaluation:**
The reviewer asked about stability, and we showed that repeated searches and different design-set sizes all lead to similar performance, indicating that BOAD behaves reliably across runs.

**Human-in-the-loop comparison:**
They mentioned that a human-in-the-loop variant could be interesting, and we agreed, while this paper focuses on full automation, such extensions are a natural direction for future exploration.

---

### Reviewer rVvm (score 6)

The reviewer found the method intuitive, appreciated the UCB-based exploration strategy, and viewed both the clarity and empirical results positively.

**Naive evolutionary search baseline:**
We implemented the requested evolutionary baseline and, under the same budget, showed that BOAD achieves higher performance while being more efficient in its use of LLM calls.

**Experimental details, cost, and variance:**
We clarified that Claude is used only during the search phase and that all methods rely on the same model at test time, and we provided both a cost comparison and a variance check demonstrating the stability of our reported results.

**Novelty of discovered sub-agents and meta-prompts:**
We confirmed that BOAD discovers sub-agents beyond the examples listed in the meta-prompts and added the complete prompt sets for all methods in the appendix

---

> ### Author Response · Authors · 2025-12-01
>
> ### Reviewer xwJx (score 8)
>
> This reviewer was very positive about the novelty of framing multi-agent design as a bandit problem and the great improvements on SWE-Bench Verified and Live.
>
> **Extending beyond prompts to broader agent design:**
> The reviewer asked whether BOAD can extend beyond prompt optimization. We clarified that BOAD is agnostic to what is being optimized and can naturally apply to broader design choices, including scaffolding.
>
> **Quantitative support for behavioral claims (Section 5.4):**
> The reviewer asked for quantitative evidence supporting our qualitative observations. We added patch-level statistics on SWE-Bench-Live showing that the BOAD-discovered system produces more focused and concise edits than the single-agent baseline.
>
> **Characterizing discovered agents and their trajectory:**
> The reviewer asked for a clearer picture of how sub-agents evolve over time. We added a chronological summary in the Appendix showing how BOAD moves from broader utilities to more specialized roles, illustrating a natural refinement process.
>
> ---
>
> In summary, the reviewers were enthusiastic about the core ideas and empirical results, and we addressed their remaining questions with additional experiments, clearer definitions, and expanded qualitative and quantitative analysis.
>
> Thank you again for overseeing this process and for considering our submission.

---

### Meta-Review · Area_Chair_yLhc · 2026-01-05

**Summary:**

The paper initially received scores of **4/6/6/8**. During the discussion, **Reviewer 5qkT** (the initial 4) indicated they would raise their score to **6**. Overall, the inclination across reviews was positive, and the authors addressed the main concerns.

The reviewers’ concerns mainly fell into three categories:
1. **Methodological grounding and baselines**, raised by **Reviewer 5qkT** and **Reviewer rVvm** (explicit request for an evolutionary-search baseline), and also by **Reviewer 5qkT** (clearer definition and stronger support for "efficiency");
2. **Experimental transparency and robustness**, raised by **Reviewer Mz7z** (insufficient details of the optimization process/final system; robustness beyond a single run), **Reviewer rVvm** (cost/fairness reporting; evaluation noise/determinism), and **Reviewer 5qkT** (sensitivity to design-set size);
3. **Scope and interpretability**, raised mainly by **Reviewer xwJx** (extension beyond prompt optimization; more quantitative behavioral analysis; clearer exposition of discovered agents/trajectories), and also by **Reviewer Mz7z** (clearer characterization of the discovered system and why the selected sub-agents help).

The authors addressed most concerns in the rebuttal, supporting an accept recommendation.

**Reviewer Concerns:**

Addressed by the rebuttal
- **Design-set size / sensitivity (5qkT):** The authors clarified the provenance of the 12-issue design set (one per SWE-bench-Verified repo) and added an ablation over design-set size showing limited sensitivity.
- **Missing evolutionary-search baseline (5qkT, rVvm):** The authors implemented an evolutionary baseline and reported that BOAD outperformed evolution on SWE-bench-Live under comparable settings.
- **Lack of clarity on "efficiency" (5qkT):** The authors clarified that efficiency refers to search efficiency and stated that they would strengthen the empirical support in the final version.
- **Insufficient details on the optimization process and final discovered system (Mz7z):** The authors clarified the final sub-agent composition, and added prompts plus the discovered sub-agent list to the appendix.
- **Robustness to stochasticity / single-run concerns (Mz7z):** The authors reported an additional run consistent with the original outcome and added further robustness evidence by providing an additional design-set-size ablation.
- **Fairness and missing experimental details (rVvm):** The authors clarified the role of Claude (search only), ensured test-time comparisons used the same model, and added cost reporting and repeated-run evidence to support stability/fairness.
- **Prompt transparency and novelty under long meta-prompts (rVvm):** The authors added complete prompt sets and argued that discovered sub-agents go beyond the examples enumerated in the meta-prompts.
- **Scope beyond prompt content + interpretability (xwJx):** The authors argued the framework is agnostic to what is optimized (potentially extending to scaffolds), added quantitative behavioral evidence (patch statistics), and provided clearer characterization of discovered agents and trajectories.


Still outstanding or partially outstanding
- **Human-in-the-loop baseline (Mz7z):** The authors acknowledged the value of a human-guided baseline but deferred it to future work. This point is not critical for the paper’s main claims, though such a comparison could be informative in follow-up work.
- **"Overkill" concern / necessity of search (Mz7z):** The reviewer questioned whether the proposed search procedure is necessary given that the final solution selects only a small number of seemingly intuitive sub-agents. While the authors provided a rationale, the argument would be stronger with additional quantitative evidence (e.g., how often plausible candidates fail).
- **Empirical demonstration of extension beyond prompts to scaffolding/system design (xwJx):** The authors argued extensibility conceptually, but an explicit experiment optimizing scaffolding (not just prompt content) would further solidify this point.

**Reviewer Scores:**

- **Reviewer 5qkT: 4 to 6**. The key concerns were addressed, and the reviewer indicated they were satisfied and willing to increase their score to 6.
- **Reviewer Mz7z:** **6 (no change).** The authors addressed requests for more detail  and robustness. The remaining items (human-in-the-loop baseline; "overkill" framing) were non-central and unlikely to shift the score materially upward or downward.
- **Reviewer rVvm:** **6 (no change).** The authors directly addressed the baseline and experimental fairness/detail concerns. Therefore, the most conservative expectation is that the score remains at 6.
- **Reviewer xwJx:** **8 (no change).** The reviewer was strongly positive. The additional  quantitative evidence, clearer description of discovered sub-agents and trajectories, and discussion of extensibility support maintaining the high score.

---

### Decision · Program_Chairs · 2026-01-26

Accept (Poster)